# Multivariate extremes in lakes

R. Iestyn Woolway [1] ✉, Yan Tong [2], Lian Feng [2], Gang Zhao [3], Dieu Anh Dinh [4], Haoran Shi [1], Yunlin Zhang [5,6] & Kun Shi [5,6]

Extreme within-lake conditions have the potential to exert detrimental effects on lakes. Here we use satellite observations to investigate how the occurrence of multiple types of extremes, notably algal blooms, lake heatwaves, and low lake levels, have varied in 2724 lakes since the 1980s. Our study, which focuses on bloom-affected lakes, suggests that 75% of studied lakes have experienced a concurrent increase in at least two of the extremes considered (27% defined as having a notable increase), with 25% experiencing an increase in frequency of all three extremes (5% had a notable increase). The greatest increases in the frequency of these extremes were found in regions that have experienced increases in agricultural fertilizer use, lake warming, and a decline in water availability. As extremes in lakes become more common, understanding their impacts must be a primary focus of future studies and they must be carefully considered in future risk assessments.

The occurrence of extreme events in lakes is becoming more apparent, with recent evidence suggesting an increase in, among others, the frequency of algal blooms[1–3], lake heatwaves[4] and anomalously low lake levels[5–8]. These within-lake extremes can have a dramatic influence on the functioning of aquatic ecosystems as well as result in numerous negative impacts on the many ecosystem services and benefits that they provide to society. For example, algal blooms, which have increased in many, but not all[3,9], studied lakes in recent decades, are among some of the main causes of poor water quality and can lead to serious health issues[10,11]. Lake heatwaves, defined as periods of extreme warm lake surface temperature, can expose aquatic organisms to oftentimes lethal conditions, leading to a risk of mass mortality events[12,13]. Moreover, while natural fluctuations in lake level and surface extent are important for aquatic ecosystems, extreme water level declines not only reduce access to freshwater for local communities which depend on them, but can also affect multiple physical, chemical, and biological lake processes[14,15].

Previous studies that have investigated extreme events in lakes have typically focused on the occurrence of univariate extremes, such as lake heatwaves or algal blooms, specifically exploring how their discrete frequency has changed through time[3,4]. However, an emerging concern for lake ecosystems is the increased occurrence of bivariate (two) or multivariate (three) extremes, i.e., situations where the frequency of more than one extreme event are increasing simultaneously. The importance of bivariate and multivariate extremes in lakes is widely acknowledged, particularly given that the abrupt nature of these events can rapidly push lake ecosystems beyond the limits of their resilience. Indeed, multivariate extreme events have the potential to exacerbate negative impacts compared to univariate extremes, leading to more severe ecologically and socioeconomically damaging events. Evaluating multivariate extreme events in lakes is thus essential for quantifying the impact of environmental and climate change on aquatic ecosystem.

In this study, we provide a global assessment of changes in univariate, bivariate and multivariate extreme events in lakes, namely in the occurrence of algal blooms, lake heatwaves, and anomalously low lake levels. To investigate these changes, we compiled data from the scientific literature, specifically those available from three key data sources (see Methods for further details): (i) the global bloom dataset (GBD) of ref. 3, which includes remotely sensed information on the occurrence of algal blooms in lakes, covering three distinct decadal periods (1982-1999, 2000-2009, and 2010-2019); (ii) the global lake

---

[1]School of Ocean Sciences, Bangor University, Anglesey, Wales, UK. [2]School of Environmental Science and Engineering, Southern University of Science and Technology, Shenzhen, China. [3]Key Laboratory of Water Cycle and Related Land Surface Processes, Institute of Geographic Sciences and Natural Resources Research, Chinese Academy of Sciences, Beijing, China. [4]Centre for Freshwater and Environmental Studies, Dundalk Institute of Technology, Dundalk, Ireland. [5]Taihu Laboratory for Lake Ecosystem Research, State Key Laboratory of Lake Science and Environment, Nanjing Institute of Geography and Limnology, Chinese Academy of Sciences, Nanjing 210008, China. [6]University of Chinese Academy of Sciences, Beijing 100049, China. ✉e-mail: iestyn.woolway@bangor.ac.uk

surface water temperature dataset (GLAST) of ref. [16], which offers daily information on the near-surface temperature of lakes worldwide during the period of 1981-2020; and (iii) the Global Lake Evaporation Volume (GLEV) dataset, which comprises historical observations of global lake surface water extent at a monthly temporal scale[17]. We identified 2724 lakes that had information available from the three datasets described above (see Methods). Using the available data, we then investigated changes in the occurrence of extreme events in bloom-affected lakes worldwide and calculated, for each site, the occurrence of algal blooms, lake heatwaves, and extreme low water extent (see Methods). Subsequently, we calculated decadal changes in the occurrence frequency of these extremes between two time periods of interest, namely the historic (-1980s to 1999) and contemporary (2010 to 2019) period.

## Results

### Variations in univariate extreme events

We calculated multi-decadal changes in the occurrence of univariate extreme events in the 2724 studied lakes (Supplementary Data 1). It is important to reiterate that in this study we focus solely on bloom-affected lakes, defined according to ref. [3]. as those that have experienced an algal bloom during the historic period. Approximately 90% of the lakes investigated by ref. [3]. do not experience algal blooms. Across the studied sites, our data suggests that, between the two time periods of interest, the frequency of algal blooms have increased (> 0%) in 56% of lakes, by $2.7 \pm 7.9\%$ on average (Fig. 1). The summary statistics quoted here, and elsewhere in the manuscript, represent the mean and standard deviation across the studied lakes. Between the two time periods of interest, a Kolmogorov-Smirnov (K-S) test suggested that the change in algal bloom frequency was statistically significant (Supplementary Fig. 1). Moreover, in this study, we classified the studied lakes as experiencing a notable (or considerable) increase or decrease in the occurrence of algal blooms if their decadal change in frequency surpassed a 0.4 increase in relative frequency (that closely aligned with a statistically significant change (see Methods)). We calculate that 40% of the studied lakes experienced a notable increase in algal blooms frequency and 26% experienced a notable decrease. Based on this classification, one could thus estimate that 34% of lakes have not experienced a notable increase/decrease and could be categorized as stable. Our analysis also suggests that multi-decadal alterations in the frequency of algal blooms can be considerably greater in some lakes or in specific regions (Fig. 1).

By ranking the observed alterations in algal bloom frequency among the studied lakes, we find that approximately one-third (number of lakes [$n$] = 37) of the top 100 (i.e., the 100 lakes with the highest increase in bloom frequency) are situated in India and approximately one-quarter ($n = 28$) are in China. In these two countries the frequency of algal blooms has increased on average, i.e., across all lakes, by $16.1 \pm 14.0\%$ ($n = 145$) and $7.1 \pm 9.2\%$ ($n = 404$), respectively (Supplementary Table 1). These results largely align with our expectations given documented increases in agricultural fertilizer consumption in both countries in recent decades (Supplementary Fig. 2; Supplementary Table 2). Specifically, one of the dominant drivers of change in the frequency of algal blooms in lakes is nutrient enrichment, primarily via Nitrogen (N) and Phosphorus (P) from agricultural fertilizer. In India, N and P fertilizer use has increased by 52 kg ha$^{-1}$ and 24 kg ha$^{-1}$, respectively, between the historic and contemporary period. In China, the increase is even higher, with N and P fertilizer use having increased by 69 kg ha$^{-1}$ and 44 kg ha$^{-1}$, respectively. The increase in fertilizer use in these countries is among the highest in the world. More broadly, N and P fertilizer use has increased in Asia, on average, by 39 kg ha$^{-1}$ and 18 kg ha$^{-1}$, respectively, during the study period (Supplementary Table 2). This has likely contributed to the $8.2 \pm 11.2\%$ average increase in algal bloom frequency in Asia since the 1980s. Moreover, the increase in algal bloom frequency is almost an order of magnitude

greater than that estimated in any other continent (Supplementary Table 4). Specifically, in Europe, where nearly one-quarter of the studied lakes are located ($n = 1040$), the average change in algal bloom frequency is $1.0 \pm 6.1\%$ since the 1980s, which could be the result of a documented decrease in N and P fertilizer use of −7 kg ha$^{-1}$ and −16 kg ha$^{-1}$, respectively, across the continent during the same period (Supplementary Fig. 3). Other factors, such as an increase in surface water temperature (see below), likely contributed to the marginal average increase in algal bloom frequency in Europe despite the overall decline in the application of N and P fertilizer (Fig. 1). It is also important to note that even when external loading of nutrients have declined, internal loading from lake sediments may also lead to more frequent algal blooms; this could also explain the marginal increase in algal bloom frequency in Europe[18].

Regarding the frequency of lake heatwaves, our analysis suggests that across the 2724 studied sites, their frequency has increased (i.e., > 0%) in the vast majority (92%) of lakes, by $4.7 \pm 3.8\%$ on average (Fig. 1). However, we also calculate that only 46% of the studied lakes experienced a notable increase (i.e., where the calculated annual changes were statistically significant; see Methods), whereas 1% experienced a notable decrease. The remaining lakes (i.e., 53%) could be considered stable. A K-S test indicated that changes in the statistical distribution in the frequency of lake heatwaves between the two periods of interest were significant (Supplementary Fig. 1). The water bodies which have experienced the greatest increase include, among others, those situated in Europe (Supplementary Table 1; Supplementary Table 4). On average, lakes in Europe have experienced a $6.5 \pm 3.6\%$ increase in the frequency of lake heatwaves. This is considerably higher than the estimated increase for lakes in Asia ($3.1 \pm 3.2\%$) and higher than estimated for lakes in both North and South America ($3.3 \pm 3.2\%$ [$n = 501$] and $4.7 \pm 4.2\%$ [$n = 270$], respectively). The substantial increase in lake heatwave frequency in Europe is primarily a result of rapid lake warming across the continent during the historic to contemporary period[19,20]. Among the 2724 studied lakes, we estimate a Spearman's rank correlation coefficient of 0.35 between the change in mean lake surface temperature and the frequency of lake heatwaves. On average, we estimate that lakes in Europe have warmed by $0.43 \pm 0.26$ °C during the study period, which is higher than the estimated change for lakes in Asia ($0.36 \pm 0.21$ °C) and greater than the estimated increase for lakes in North ($0.25 \pm 0.19$ °C) and South America ($0.30 \pm 0.14$ °C). It is also important to consider the role that other factors, besides mean surface warming, play in the occurrence frequency of lake heatwaves. Specifically, the climatological seasonal cycle of lake surface temperature is a key factor influencing lake heatwave alterations within a warming world[4]. Notably, lakes that experience low seasonal temperature variability (e.g., tropical lakes) can experience more frequent lake heatwaves under relatively minimal warming, as the lake heatwave threshold can be more easily exceeded. Our analysis supports this assertion. For example, African lakes ($n = 99$), which typically experience minimal seasonal temperature variability[21], have experienced a $5.3 \pm 3.6\%$ increase in the frequency of lake heatwaves since the 1980s, one of the highest in our study, despite only warming by $0.30 \pm 0.15$ °C, the second lowest in our study (Supplementary Table 1; Supplementary Table 4).

In terms of low water level extremes, our data suggests that, between the two time periods of interest, their occurrence frequency has increased (>0%) in 50% of the 2724 studied lakes, by $0.7 \pm 13.6\%$ on average (Fig. 1). Our analysis indicated that these changes likely did not arise due to random variability and that the empirical distribution functions of low water extremes from the two epochs differed significantly from each other (Supplementary Fig. 1). We also calculate that 18% of the studied lakes experienced a notable increase (i.e., where the calculated annual changes were statistically significant; see Methods) and 20% experienced a notable decrease. In turn, 62% of lakes could be considered 'stable'. For those lakes that have experienced an

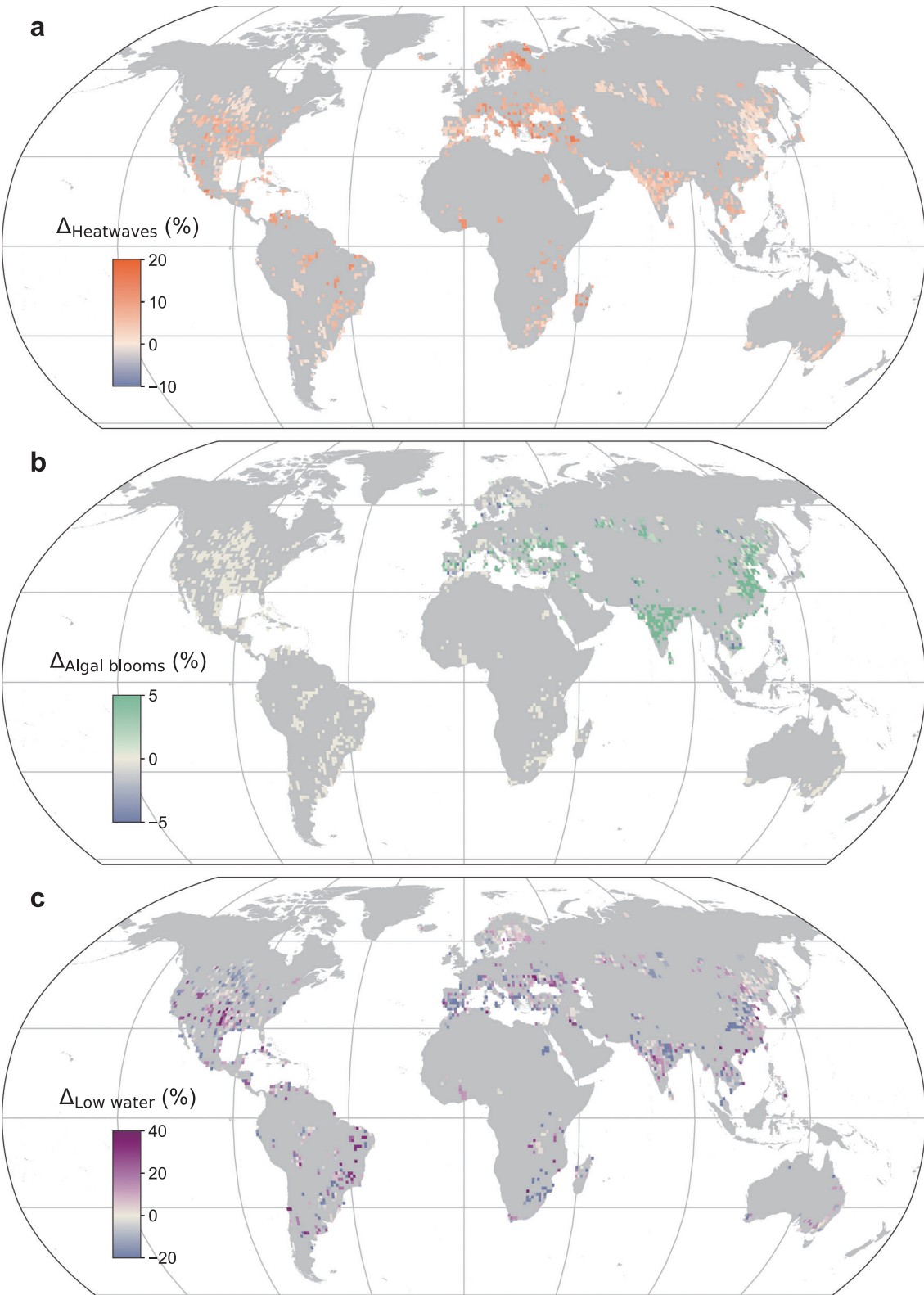

**Fig. 1 | Change in the occurrence frequency of univariate extreme events in lakes.** Shown are the estimated changes in the frequency of (**a**) lake heatwaves, (**b**) algal blooms, and (**c**) anomalously low water extent in 2724 globally distributed lakes between the historic (-1980s to 1999) and contemporary (2010 to 2019) period. Negative values suggest that the frequency of a specific extreme event has decreased during the study period. Data are aggregated into 1° × 1° grid cells. Source data are provided as a Source Data file.

increase in the frequency of this extreme event, we calculate an average increase of 11.7 ± 8.8%. Our data suggest that approximately one-quarter of the top 100 ranked lakes (i.e., in terms of the change in low water extremes) are situated in the United States. However, if we consider all studied lakes from the United States ($n = 328$), the average change in the frequency of low water extremes is relatively small (= 0.11 ± 15.5%). Our study also highlights regions that have experienced a considerable increase in low water extremes, including southern Australia, northern China, and central Asia, as reported in previous studies[7]. An increase in the frequency of low water extremes in lakes is primarily influenced by changes in mean surface extent between the historic and contemporary period. Indeed, among the 2724 studied lakes, we estimate a Spearman's rank correlation coefficient of −0.81 between the percent change in mean lake water extent and the frequency of low water extremes. Thus, lakes that experience the greatest relative decline in mean surface extent also typically experience the greatest increase in the frequency of low water extremes. The primary factors that influence mean water extent and, in turn, the frequency of low water extremes in lakes include changes in flow regime (i.e., streamflow) that are linked to global climate change, the volume of water withdrawn for agricultural and domestic purposes, and the volume of water lost via evaporation, all of which have changed in recent decades[17,22–24] (Supplementary Fig. 3). The two dominant drivers of change in the occurrence of low water extremes include streamflow and water use. Across the studied sites, we observe that positive changes in low water extremes are associated with larger decrease of streamflow and larger increase of water use (Supplementary Fig. 4). Such patterns are likely to persist in the future in many regions in the context of population growth and the enhancement of agricultural and atmospheric water demands[25]. In addition, the volume of water lost via evaporation can influence the occurrence of low water extremes. However, from a volumetric perspective, lake evaporation changes are much smaller than the changes in streamflow and water use and is not a dominant factor for explaining changes in low water extremes in the studied lakes.

It is important to acknowledge that some lakes within our dataset exhibited marginal changes in extreme event occurrences, reported above as stable. These changes might fall within the range of uncertainties inherent to observational data. These uncertainties can arise from various sources, including measurement error and data limitations. As such, the interpretation of these marginal changes requires careful consideration. We also emphasize the need to be cautious when interpreting marginal changes, as they may not always carry ecological or practical significance. In our statistical analyses, we have incorporated measures of statistical significance to offer an assessment of the reported changes in extreme event occurrences. However, we cannot be certain that the statistical definition of extreme events in our studied lakes lead to ecological or socioeconomic impact.

**Decadal alterations in bivariate and multivariate extreme events**
Alterations in bivariate extreme events are considered as situations where the frequency of more than one of the univariate extremes are increasing simultaneously within a lake. Ultimately, here we investigate, for each pair of extreme events, how their frequency has changed concurrently between the historic and contemporary period. Specifically, we investigate what percentage of lakes with available data have experienced parallel changes in the frequency of (i) algal blooms and lake heatwaves, (ii) lake heatwaves and low water extremes, and (iii) algal blooms and low water extremes. Overall, our study suggests that approximately three quarters of the studied lakes ($n = 2035$; 75%) have experienced a simultaneous increase (27% experiencing a notable increase) in at least one of these bivariate extreme events during the study period, with some (see below) experiencing an increase in all three extremes.

Our analysis suggests that 51% of the studied lakes ($n = 1398$) have experienced a concurrent increase (20% experiencing a notable increase and 0% a notable decrease) in the frequency of algal blooms and lake heatwaves (Fig. 2). An increase in this bivariate extreme can occur when the drivers of the univariate extremes (i.e., agricultural fertilizer consumption and mean lake surface temperature) are simultaneously increasing. Given that most of the studied lakes have experienced an increase in the frequency of lake heatwaves (see above), an increase in the occurrence of this specific bivariate extreme will be influenced primarily by changes in the frequency of algal blooms. Among the studied sites, our data suggest that lakes in Asia, notably those situated in India and China, have experienced the most consistent simultaneous increase in the frequency of algal blooms and lake heatwaves (Fig. 2; Supplementary Table 5; Supplementary Table 6). Notably, we calculate that 95% and 69% of lakes with available data in India and China, respectively, have experienced an increase in the frequency of this bivariate extreme. Furthermore, 75% of the studied lakes in Asia experienced a simultaneous increase in the frequency of algal blooms and lake heatwaves, which is considerably higher than in any other continent (e.g., 37% of studied lakes in Europe).

Regarding the second bivariate extreme of interest, we estimate that 46% ($n = 1266$) of the studied lakes have experienced a simultaneous increase (9.5% experiencing a notable increase and 0% a notable decrease) in the frequency of lake heatwaves and low water extremes (Fig. 2). Among the studied sites, 55% of those situated in Europe have experienced a simultaneous increase in the frequency of lake heatwaves and low water extremes since the 1980s (Supplementary Table 6). Specifically, the greatest percentage of lakes within a country that experienced an increase in this bivariate extreme were 66% in Finland ($n = 365$) and 65% in Sweden ($n = 101$); note that we restricted our analysis to countries where at least 50 lakes had available data (Supplementary Table 5).

Our data suggests that the third bivariate extreme of interest, notably a joint increase in the frequency of algal blooms and low water extremes, changed less consistently across the studied lakes compared to the other bivariate extremes described above. Most notably, 27% of the studied lakes experienced an increase (8% with a notable increase and 3.8% with a notable decrease) in the frequency of this bivariate extreme event (Fig. 2). However, the number of lakes within a specific country or continent that experienced a change can be considerably higher. For example, 42% of the studied lakes in Asia experienced an increase in this bivariate extreme event with the studied lakes in India (54%) experiencing the most pronounced increase (Supplementary Table 5; Supplementary Table 6).

As well as experiencing an increase in the frequency of bivariate extreme events, our analysis suggests that a proportion of the studied lakes have also experienced an increase in the occurrence of multivariate extremes. We define an increase in multivariate extreme events as situations where the frequency of the three univariate extremes considered have increased simultaneously within a lake since the 1980s. Specifically, using all available information for the lakes of interest, we calculate how the frequency of multivariate extremes have changed between the historic and contemporary period. We calculate that approximately one-quarter of the studied lakes ($n = 687$; 25%) have experienced a simultaneous increase (5% with a notable increase and 0% with a notable decrease) in all three univariate extremes since the 1980s (Fig. 3). However, across some continents (e.g., Asia; 37%) and notably within some countries, the percentage of studied lakes that have experienced an increase in the frequency of multivariate extremes is much higher (Supplementary Table 6). For example, in India approximately half (52%) of the studied lakes experienced an increase in the occurrence frequency of multivariate extreme events (Supplementary Table 5). This is particularly concerning given that India is now one of the most populated countries in the world, and that its growing population is being subjected to more extreme conditions

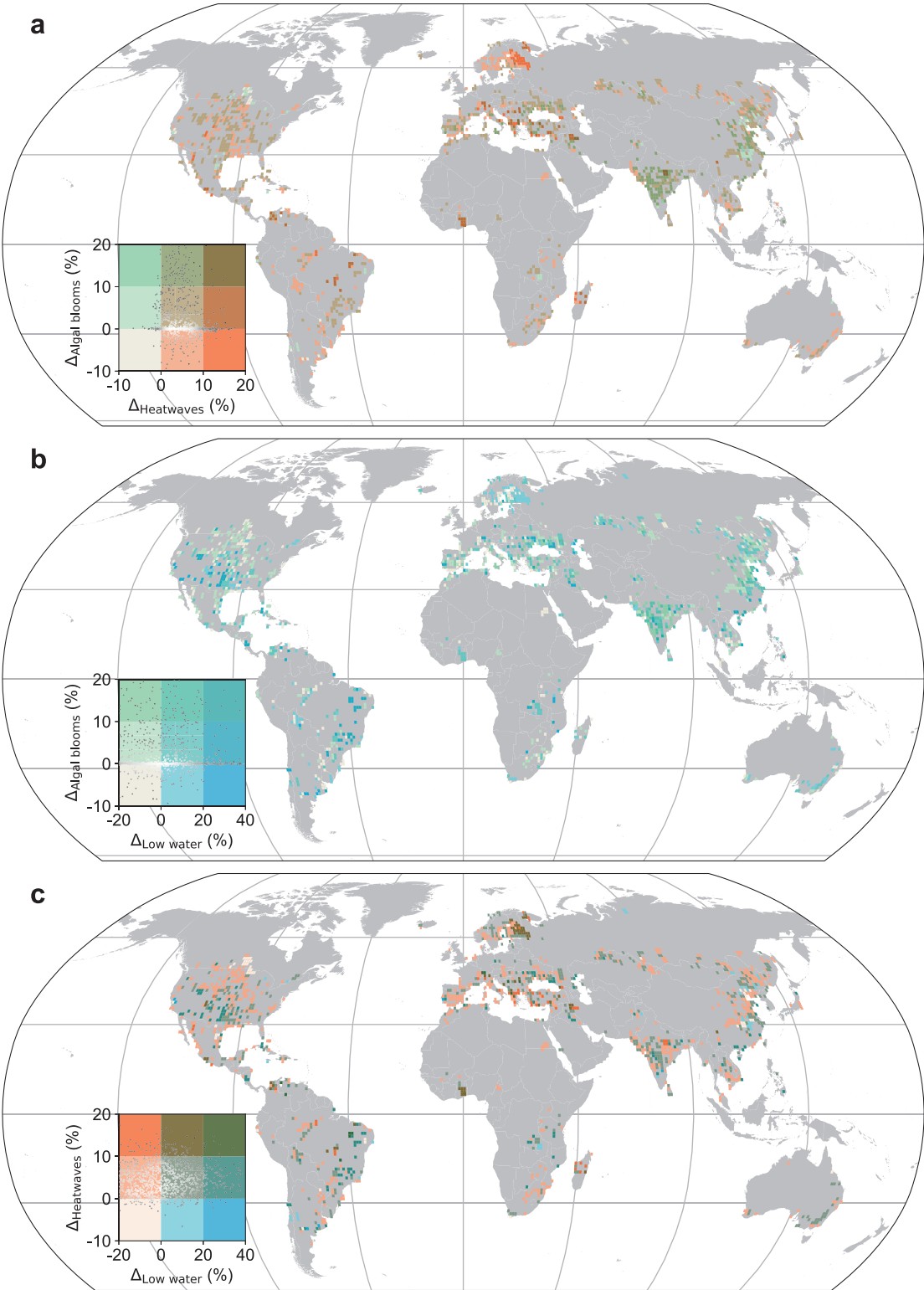

**Fig. 2 | Change in the occurrence frequency of bivariate extreme events in lakes.** Shown are the estimated changes in the frequency of bivariate extremes, considered as situations where the frequency of more than one of the univariate extremes have increased simultaneously within a lake between the historic (-1980s to 1999) and contemporary (2010 to 2019) period. We calculate concurrent changes in the occurrence frequency of (**a**) algal blooms and lake heatwaves, (**b**) algal blooms and low water extremes, and (**c**) lake heatwaves and low water extremes. Data are aggregated into 1° × 1° grid cells. Source data are provided as a Source Data file.

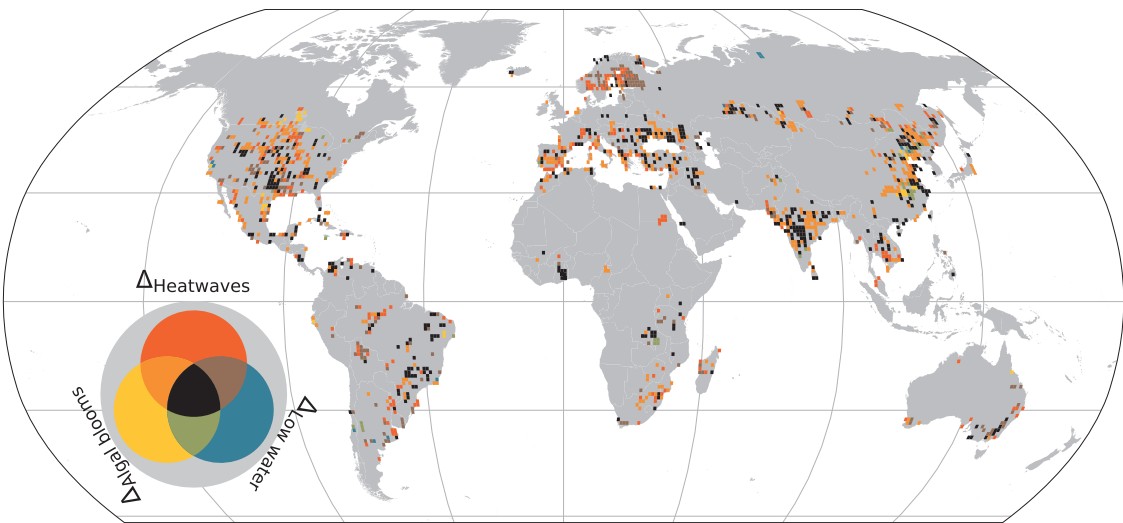

**Fig. 3 | Change in the occurrence frequency of multivariate extreme events in global lakes.** Shown are changes in the frequency of multivariate extreme events, which we define as situations where the frequency of the three extremes considered (algal blooms, lake heatwaves, and anomalously low water extent) have increased simultaneously since the 1980s. Different colors, as shown in the legend, can represent a change in frequency of either an individual or multiple extreme events. For example, solid colors represent a positive change for a specific covariate, whereas transparent colors represent a situation where the frequency of specific pairs of covariates have increased. The center piece of the legend (i.e., black color) represents a situation where all covariates follow a positive change. A gray colour suggests that all covariates experienced a negative change in terms of the occurrence frequency of each extreme. Data are aggregated into 1° × 1° grid cells. Source data are provided as a Source Data file.

regarding one of its important sources of freshwater. Given historic increases in agricultural fertilizer use, climatic warming and freshwater withdrawal, India has experienced a perfect storm of conditions that has led to an increase in the frequency of multivariate extreme events in lakes.

## Discussion

Individually, each of the extreme events described in this study can have considerable, and diverse, impacts on lake ecosystems, as well as more broadly. For example, while some littoral communities are adapted to periodic drying and flooding and even rely on these perturbations to support life history stages[26], anomalous littoral exposure that occurs during extreme low water events can degrade nearshore habitat structure, alter biotic community composition, and impact critical lake ecological functions[14,27,28]. Critically, changes in water level can directly affect the availability of habitat for various organisms, particularly those dependent on shoreline areas for breeding, foraging, or shelter. In extreme cases, dramatic shifts in water levels can even result in the loss of critical habitats and the displacement of sensitive species. A reduction in water level can also increase the amount of sunlight reaching deeper waters, leading to a change in dominant primary producers with cascading effects at higher trophic levels[29]. Furthermore, dramatic declines in lake level and surface water extent can influence human activity, with already vulnerable communities having to travel greater distances to reach freshwater[30]. It can also influence the local climate[31,32] as well as greenhouse gas emissions from lakes[33,34]. Regarding lake heatwaves, recent studies have suggested that an increase in their frequency can have a detrimental influence on aquatic organisms by exposing them to unprecedented temperatures, which can be particularly fatal for species that live in regions close to their critical thermal maximum[12,13,35]. Specifically, when temperatures more frequently exceed critical levels, organism traits such as embryonic development, growth, or heat tolerance can be severely affected. Extreme temperatures have even been described as potentially having a disproportionate impact on species relative to gradual, long-term temperature change[36]. An increase in surface water temperature during a lake heatwave can also result in other critical changes, such as a decline in dissolved oxygen concentrations at the lake surface due to reduced gas solubility and at depth due to stronger and longer lasting thermal stratification[37,38]. Finally, algal blooms can have a devastating impact on lake ecosystems. For example, blooms of algae in freshwaters can produce potent toxins, potentially making water bodies unsafe for recreational activities and threatening the health of humans, pets, or livestock that use affected waters. These blooms are often produced by cyanobacteria which can cause a range of health problems, from minor skin irritations to severe stomach upsets, and can even lead to death[39]. However, even if algal blooms are non-toxic, they can negatively impact aquatic life by, for example, irritating or even clogging fish gills causing suffocation and blocking out sunlight from reaching deeper water. The latter can reduce photosynthesis, ultimately leading to a decrease in the concentration of dissolved oxygen available to support aquatic life[38]. In addition, when algae eventually die and decompose, they consume oxygen in the water, causing dead zones that can be inhospitable for many organisms. The detrimental effects of algal blooms can ripple throughout the ecosystem, impacting not only water quality but also the availability of resources for other species, such as zooplankton and fish.

Whilst the potential impact of univariate extreme events on lake ecosystems are troubling, the impact of more frequent bivariate or multivariate extremes could arguably be much greater. Indeed, simultaneous changes in lake heatwaves, algal blooms, and lake water extent could lead to a complex interplay of environmental factors that significantly influence aquatic ecosystems. Most notably, the synergistic effects of these changes can create a cascade of ecological consequences that amplify their individual impacts. For instance, elevated temperatures not only directly encourage algal blooms but also reduce the oxygen-carrying capacity of water, exacerbating the negative effects of these blooms on aquatic life. Similarly, fluctuating water levels can interact with temperature changes to alter the timing and extent of algal blooms, further complicating the ecological dynamics. This interconnectedness underscores the need for a holistic and multidisciplinary approach to studying and managing lake ecosystems. Understanding the complex relationships between these factors and their combined effects is essential for effective conservation and mitigation strategies. Addressing the simultaneous changes in lake temperature, algal blooms, and water level is a critical step in

preserving the delicate balance of these ecosystems and ensuring their resilience in the face of ongoing environmental challenges.

An increase in the frequency of bivariate and multivariate extremes in lakes can be driven by several factors. For example, the frequency of concurrent algal blooms and lake heatwaves can increase due to changes in external drivers (e.g., fertilizer use and climatic warming), or they can also arise when the driver of one extreme, such as anomalously hot surface water temperature causing a lake heatwave, also cause other relevant changes, such as an algal bloom due to the temperature dependence of algal growth[40–42]. However, it is also important to note that a synchronous increase in the frequency of the univariate extreme events may not always occur, as one extreme could, in fact, suppress the occurrence of another. For example, previous studies have shown that higher lake surface temperatures that can lead to a lake heatwave often also result in stronger thermal stratification and, in turn, a reduced supply of nutrient rich bottom waters to the near-surface layer, leading to nutrient limitation at the surface and thus reduced phytoplankton growth[43–45].

In terms of an increase in the occurrence frequency of lake heatwaves and low water extremes; these will largely occur in lakes where the average surface water temperature has increased considerably above the climatology and where the mean surface water extent has decreased between the historic and contemporary period. This coherent change could occur, for example, when surface water temperature increases within a warming world and, at the same time, water is withdrawn from a lake for agricultural purposes. However, surface water temperature and lake water extent are also closely related where, for example, a change in one variable could instigate a change in the other. For example, lake evaporation, which is an important driver of lake level and surface water extent, can increase during periods of high surface water temperature, resulting in positive feedback between the univariate extremes. In contrast, higher evaporation rates and thus a short-term decline in surface extent could lead to cooler lake surface temperatures[16] and thus a subsequent decline in the frequency of lake heatwaves. Indeed, the relationship between lake heatwaves and low water extremes is complex and one should not expect a direct relationship across lakes worldwide, particularly given the influence of other external factors (e.g., precipitation, river inflow, groundwater flux, withdrawal) on the lake water budget.

A simultaneous increase in the frequency of algal blooms and low water extremes will most likely be influenced by a concurrent increase in the external factors that contribute to their individual occurrences. However, some studies have suggested that these lake properties are also related, e.g., a decrease in surface water extent can lead to an increased occurrence of algal blooms. Most notably, drought conditions, and a drawdown in lake level and surface extent, can result in an increase in retention time and a subsequent relative increase in water column nutrient concentrations, which can lead to algal blooms[44,46,47], as well as promote nutrient resuspension from the sediments[18]. In contrast, during periods of heavy precipitation, where a sudden pulse of water can flow into a lake, influent water can also increase the amount of nutrients entering the system, subsequently leading to algal blooms[1,48]. Often high summer rainfall events can transport a large proportion of the annual load and result in exceptional blooms. However, there is also evidence suggesting that nutrient concentrations in lakes could be reduced through greater flushing due to precipitation, leading to less algal blooms[42].

We understand relatively little about the impact of multivariate extremes, particularly in terms of their knock-on effects on the physical, biological, and chemical environment of lakes. Indeed, it is challenging to anticipate the precise effects due to the interactions that occur within a water body and how these differ globally. The complex interplay of selective pressures due to multivariate extremes makes emergent effects on lake ecosystems difficult to predict and underlines the need for empirical data and detailed modeling

approaches that can accurately capture such changes, whether it be via process-based modeling, mesocosm experiments, or large-scale meta-analyses. In addition, studies investigating additive, antagonistic and synergistic effects are critical to fill this knowledge gap[49–51]. Understanding how various extreme factors interact and influence each other is critical for forecasting and early warning systems. Advanced modeling approaches, informed by real-time data and artificial intelligence, can enhance our ability to anticipate and respond to extreme events. Overall, the effects of multivariate extremes are likely to cascade through whole ecosystems, but we currently lack critical knowledge of the direction and magnitude of these effects. It is also important to acknowledge that we do not currently know which of the impacts described above will dominate as multivariate extreme events become more frequent. Certain types of bi- or multi-variate extremes could have opposite impacts on lake ecosystems, such as low water levels and algal blooms, respectively, increasing or reducing light availability in deeper water. Other bivariate extremes could work synergistically to have an even greater detrimental impact, such as lake heatwaves and algal blooms, leading to widespread deoxygenation. These potential impacts should be a primary focus of future work. It is also imperative that we investigate the spatial-temporal relationships between the extremes and, when data becomes available, delve into the underlying drivers of concurrent increases in different extremes as well as the occurrence of compounding events, where two or more extremes occur simultaneously or sequentially as well as their drivers[52–55].

An increase in the frequency of bivariate and multivariate extreme events in our natural environments are indicative of the intricate web of challenges posed by climate change and other human-induced factors. These compound extremes are not just isolated incidents but rather the manifestation of a changing climate that demands our utmost attention and concerted efforts in mitigation and adaptation. One notable concern is the interconnectedness of these extreme events with broader environmental, societal, and economic systems. When we examine the repercussions of bivariate and multivariate extremes, we must not limit our perspective solely to the immediate ecological impacts. Instead, we should consider their ripple effects across various domains. For example, a simultaneous occurrence of prolonged drought could not only devastate local ecosystems but also threaten water supplies, agricultural production, and human settlements. These cascading effects underscore the urgent need for holistic, cross-disciplinary approaches to address and adapt to these challenges. Furthermore, bivariate and multivariate extremes highlight the limitations of traditional risk assessment and management strategies. Conventional risk assessments often focus on single hazards in isolation, but these compound events demand a shift toward a more integrative and systemic approach. Incorporating the potential interactions between multiple stressors in our risk assessments can provide a more accurate picture of the true vulnerabilities of ecosystems, infrastructure, and communities. In addition, the emergence of bivariate and multivariate extremes underscores the importance of anticipatory governance and resilience-building. Instead of reacting to disasters as they occur, we must proactively invest in adaptive strategies that can withstand and recover from these complex events. This includes designing resilient infrastructure, implementing sustainable land-use practices, and fostering community preparedness and collaboration.

There is increasing appreciation of the influence of extreme events in lakes and concerns over their effects on aquatic ecosystem functioning. Of particular concern is that an increased occurrence of these extremes could amplify the effects of gradual long-term change. However, most research to date has focused on the impacts of univariate extremes, and their influence on the aquatic ecosystem. Far fewer assessments of bivariate and multivariate extremes exist. It is critical that we accelerate knowledge of concurrent physical and

biogeochemical extreme events in lakes, which is fundamental for adaptation, planning and future risk assessments. Indeed, whilst emerging challenges for lake ecosystems are already underway, these will likely accelerate in the future, resulting in unprecedented and unpredictable changes to aquatic ecosystems. Improving our understanding of the complex impacts of multivariate extreme events in lakes is urgently needed. The rise in bivariate and multivariate extremes is not merely a coincidence but a poignant reflection of the changing dynamics of our planet. It necessitates a paradigm shift in how we perceive and manage risks, requiring interdisciplinary collaboration, anticipatory governance, and cutting-edge scientific approaches. Only through this comprehensive approach can we hope to address the multifaceted challenges posed by these complex extreme events and build a more resilient and sustainable future.

## Methods

### Study sites

The lakes included in this investigation were initially chosen based on the availability of lake specific information on the occurrence of algal blooms, surface water temperature, and lake surface water extent. This information was extracted from three openly available data sets, namely the global bloom dataset (GBD), the global lake surface water temperature dataset (GLAST), and the Global Lake Evaporation Volume (GLEV) dataset, each of which are described below. While the information on some of these variables are available for up to 1.4 million lakes worldwide (i.e., in the case of the GLEV dataset), in this study we only selected lakes where information for all three variables were available for a specific lake. Additionally, we excluded lakes with marginal historic extreme occurrence frequency of algal blooms, particularly those less than the propagated uncertainty from the limited Landsat-derived algal bloom observations during the 1980s-1999 compared to the 2010s (see below). Overall, 2724 globally distributed lakes were chosen from these datasets and, in turn, are included in this investigation (Supplementary Data 1, Supplementary Fig. 5). These lakes span various latitudes and longitudes (Supplementary Fig. 5c-f), and their distribution across different climate types closely mirrors that of global bloom-affected lakes (Supplementary Fig. 5g-h). The areas of the selected lakes range from 0.16 to 17,444.01 km², with a median size of 6.96 km². The number of small, medium, large, and large lakes accounts for 1.5% (≤1 km²), 58.4% (1–10 km²), 32.1% (10–100 km²), and 8.0% (>100 km²) respectively.

### Water body surface area and low water extremes

In this study, we opted to utilize surface water extent data obtained through remote sensing observations as a basis for our assessment of changes in lake water availability and their associated extremes. This choice was driven by practical considerations, primarily centered on the broader spatial coverage afforded by surface water extent data compared to water storage information or indeed traditional in-situ water level measurements. While in-situ measurements provide valuable information, they are typically limited to specific monitoring locations and may not offer a comprehensive view of larger lake systems or entire regions. Surface water extent data, derived from satellite observations, allow us to encompass substantial lake areas and capture variations in lake surface extent, which are often indicative of extreme events such as low water levels. To calculate the surface area of the studied lakes, and subsequently the frequency of low water extremes, we analysed information available from the GLEV dataset[17]. The GLEV dataset stands as the most high-temporal resolution lake surface area dataset available for 1.4 million lakes worldwide. In brief, monthly water surface area time series were reconstructed by ref. 17. based on a combination of the dynamic Landsat-based global surface water (GSW) dataset[56] and the static HydroLAKES shapefiles[57]. Specifically, for each month and each water body, the water classification map from the GSW dataset was extracted within the defined boundary

of the HydroLAKES shapefiles. However, as such water classification maps are frequently contaminated by cloud cover, cloud shadow, and sensor failure (e.g., Landsat 7 SLC-off data), leading to large data gaps. Ref. 58. adopted an automatic image enhancement algorithm, which detects the observable water edge and extends it to the contaminated area according to the water occurrence image, to create the complete water surface. The surface area time series at a monthly time step from 1985 to 2019 for each lake was subsequently constructed. Using the monthly varying surface area time series, we defined periods of low water extremes in each lake as when the surface water extent decreased below a monthly varying 10th percentile threshold. In turn, this approach can be used to identify anomalously low water events at any time of the year.

To assess the accuracy and reliability of our methodology in capturing variations in lake surface area, we conducted a comprehensive evaluation that involved comparisons with daily in-situ lake level measurements where available. We validated the monthly water surface area data using observed water surface elevation from 155 lakes worldwide (Supplementary Fig. 6; Supplementary Table 7). The observed elevation data were compiled from multiple water management agencies and remote sensing datasets including the Bureau of Meteorology in Australia (http://www.bom.gov.au/; $n = 15$ lakes), Central Water Commission in India (https://cwc.gov.in/; $n = 8$ lakes), US Army Corps of Engineers[59] ($n = 92$ lakes), and DAHITI[60] ($n = 40$ lakes). The median correlation coefficient is 0.76, representing the good performance of our surface area dataset in representing water dynamics. In addition, we validated the calculated extreme occurrence using the observed elevation data. Among the 155 lakes, there are 25 lakes that have complete in-situ records from 1985 to 2019 (Supplementary Fig. 7). Thus, we compared the extreme occurrence changes (from historic to contemporary period) calculated based on our area dataset and that calculated based on the observed elevation data.

To explain some of the long-term changes in water level across the studied lakes, we downloaded global scale data that represented alterations in (i) the volume of inflowing water to each lake, (ii) changes to the volume of water withdrawal from the contributing basin, and (iii) changes in evaporation volume. Data on the naturalized streamflow between the historic (1985 to 1999) and contemporary (2010 to 2019) period were retrieved from the GloFAS-ERA5 dataset with a spatial resolution of 0.1° (ref. 22). Long-term trend of total water (surface water and groundwater) withdrawal from 1971 to 2010 were retrieved from the reconstructed gridded (0.5° spatial resolution) water withdraw dataset[61]. Changes in evaporation volume were extracted from the GLEV dataset[17]. To investigate the relationship between the occurrence of low water extremes and changes in streamflow, water withdrawal and evaporation volume, we (i) aggregated the lakes into Hydro-Basin_level03 basins[57], (ii) calculated the change in low water level at basin scale using lake area as the weight; (iii) calculated the relative changes in streamflow, water use, and lake evaporation between the two periods (1980s-1999 vs. 2010-2019), and (iv) plot the distribution of the relative changes in explanatory variables against positive and negative low water extreme changes (Supplementary Fig. 4).

### Lake surface water temperature and lake heatwaves

The GLAST (Global LAke Surface water Temperature) dataset presents a comprehensive collection of lake surface water temperatures for 92,245 lakes worldwide from 1981 to 2020[16]. The development of this dataset involved the integration of satellite remote sensing and numerical modeling and was conducted in four steps. Firstly, the examined lakes were determined using permanent water surfaces within the HydroLAKES boundary. Secondly, Landsat satellite observations of lake surface water temperature were retrieved for these selected lakes using a statistical mono-window algorithm[62]. The high spatial resolution of Landsat images (60-120 m) allowed for the effective remote sensing of surface temperature of water located at

least 3 pixels away from the adjacent land[16,63]. Thirdly, four decades of Landsat derived lake surface water temperatures were used to tune the Freshwater Lake model, FLake[64,65], to determine optimal model settings at an hourly temporal scale for each water body. FLake was driven by hourly ERA5-Land climatic data[66], in line with the instantaneous nature of Landsat retrievals. The meteorological variables required to drive FLake include air temperature at 2 m, wind speed at 10 m, surface solar and thermal radiation, atmospheric pressure, and specific humidity. Times series of these parameters at each lake center were extracted from ERA5-Land at an hourly time step and at a longitude-latitude grid resolution of 0.1° × 0.1°. Using high spatial resolution satellite observations as the boundary condition for numerical simulations has proven to be an effective lake-specific calibration approach in compensating for various sources of biases, including the limitations arising from the use of coarse spatial resolution atmospheric dataset for small lakes[16]. Lastly, the FLake model with optimal model settings for each lake were conducted to simulate daily lake surface water temperature for each water body. The GLAST dataset has been validated using in-situ water temperature records across globally distributed and various types (large/small, deep/shallow, cold/temperate) of lakes, with an overall median absolute error (MAE) of 1.16 °C at the daily scale[16]. Using the daily lake surface water temperatures from the GLAST dataset, we estimated the frequency of lake heatwaves following the methods described by ref. 4. Specifically, using the R package 'heatwaveR'[67], lake heatwaves were defined as when daily lake surface temperatures were above a local and seasonally varying 90th percentile threshold for at least five consecutive days. The frequency of lake heatwaves was calculated relative to the duration of the open-water (i.e., ice-free) season in each lake.

To evaluate the ability of the GLAST dataset in capturing lake heatwave events, we conducted validation and sensitivity analysis. Instead of relying on remote sensing products (e.g., MODIS LST and ESA CCI Lakes), which suffer from data gaps and limited representation of satellite overpass moments, we chose a continuously observed in situ dataset. In total, we compiled 72,169 daily lake surface water temperature records from 17 lakes across 8 countries (Supplementary Table 8). In-situ lake surface temperature observations were compiled from the National Data Buoy Center (https://www.ndbc.noaa.gov/), King County government website (https://green2.kingcounty.gov/), North Temperate Lakes Long Term Ecological Research Network (NTL LTER) (https://lter.limnology.wisc.edu/), Swedish Infrastructure for Ecosystem Science (SITES) (https://data.fieldsites.se/), NERC Environmental Information Data Centre (https://eip.ceh.ac.uk/data), EDP (Environmental Data Portal, https://envdata.boprc.govt.nz/), Government of Ireland (https://data.gov.ie/), as well as from the literature[68], and personal communications (see Supplementary Table 8). Validation results revealed that GLAST data performed well in capturing both annual heatwave frequencies (slope: 0.92, R: 0.93) and decadal changes (Slope: 0.96, R: 0.94) among the surveyed small or large lakes (Supplementary Fig. 8).

Additionally, we conducted a sensitivity analysis for all examined lakes to assess the robustness of decadal lake heatwave frequency changes between the two focused periods (~1980s-1999 and 2010s), considering bias in the daily GLAST dataset. Random noises, matching the biases of daily FLake simulations[16], were added to the daily simulated LSWT time series dataset for each lake. The bias-added time series were used to estimate lake heatwave frequency and decadal changes. The analysis demonstrated consistent heatwave frequency occurrences and changes between the noise-added and original time series data (Supplementary Fig. 9). During the historical period, the linear slope and correlation coefficient (R) were 0.89 and 0.97, respectively. In the contemporary period, they were 0.81 and 0.96, respectively. The consistent heatwave frequencies in each period led to uniform decadal frequency changes (Slope: 0.71, R: 0.91) between the two periods. Overall, 92% of lakes exhibited the same directional

changes (increase or decrease) in heatwave frequency detected by both noise-added and original lake surface water temperature data (Supplementary Fig. 9). This suggests a small impact of temperature error propagation on the variability of heatwave frequency changes, ensuring the robustness of the computational results presented in this study. Furthermore, we conducted sensitivity tests of optimization of Flake models by excluding partial satellite observations to unveil the influence of coarse temporal resolution remote sensing data on lake temperature simulations and, consequently, on the final heatwave change detections. Our results revealed consistent validation performance and frequency changes between the models trained on three seasons or proportional data and those trained on all available seasons (Supplementary Fig. 10). Consequently, the reduced data availability presented a limited impact on the accuracy of the FLake model and the detection results for heatwave changes.

## Algal blooms
The Global Bloom Database (GBD) presents a comprehensive analysis of global freshwater algal blooms using the complete archive of 2.91 million Landsat satellite images, with 30-m resolution and 16-day revisit time, from 1982 to 2019[3]. The database represents a global characterization of algal blooms in freshwater lakes. A color-based algorithm was used to identify algal blooms from each available images, initially applied to 248,243 lakes with surface areas exceeding 0.1 km². The selection of these lakes was based on the necessity for valid satellite observations and their theoretical suitability for phytoplankton growth, determined specifically by summer mean temperature[3]. Due to the prolonged revisit cycle of Landsat and frequent disturbances such as cloud contamination, detected algal bloom events exhibit a coarse temporal resolution distribution. Multiple-year statistics were conducted to ensure the validity of detecting decadal changes in bloom occurrence. Here, bloom occurrence is defined as the ratio between the number of positive algal bloom area values and the total number of valid observations. To this end, the GBD includes information on the bloom occurrence (BO), i.e., the frequency of algal bloom detection, for three periods (1980-1990s, 2000s, and 2010s) for 21,878 affected lakes (constituting 8.8% of the initially examined lakes)[3]. Algal blooms are considered in this study as extreme events of lake eutrophication, and we do not categorize events according to their severity.

It is important to acknowledge that our study focused on a subset of lakes drawn from a specific remotely sensed database[3]. While our analysis revealed that over half of our studied lakes exhibited an increase in algal bloom frequency, we recognize that the prevalence and dynamics of algal blooms can vary significantly by region. One notable limitation of our study is the exclusion of some lakes in high-latitude regions such as Canada, Siberia, and high-mountain Asia, which are known hotspots for high lake density[69]. This exclusion was made due to data availability constraints, notably the overlap of lakes with available data from the lake algal bloom and lake temperature databases. As a result, our sample may not capture the full spectrum of algal bloom dynamics globally. The regional variations in algal bloom occurrences are indicative of the complex interplay of local environmental factors and human activities, including nutrient loading and climate patterns. Future research could benefit from incorporating a more comprehensive dataset to account for these regional disparities and to provide a more holistic understanding of the factors influencing algal blooms on a global scale.

Another important consideration in our study is the observation frequency of satellite imagery, which has evolved over time. Specifically, the observation frequency was typically lower in the 1980s and 1990s compared to the more recent decade. We acknowledge that higher observation frequency is more likely to capture extreme events in a more comprehensive manner. However, we also note that the frequency of such events, which are defined relative to the total number of days with available data, should not be considerably

impacted. A commonly used practice is to extend the observation period to acquire substantial data from multiple years for calculating valid average levels. This practice has been commonly used in many previous studies to support the robustness of change detections based on remotely sensed data with data gaps[3,7]. Nevertheless, we conducted a sensitivity analysis to assess whether the difference in observation frequency between the earlier decades and the recent decade may have influenced the reported change in algal bloom occurrence (Supplementary Figs. 11 and 12). This analysis aimed to evaluate the robustness of our findings in light of these variations. Critically, our sensitivity analysis considered the potential impact of observation frequency on the detection of extreme events. Despite the differences in observation frequency, our analysis revealed that the overall change direction (increase or decrease) of algal blooms in Lake Taihu, China, based on partial or all-available hourly-resolution GOCI images[70] remained unchanged (Supplementary Fig. 11). Using another high-resolution satellite, Aqua MODIS, derived daily algal bloom dataset for 102 Chinese lakes[9], we performed frequency calculation for historic period and frequency change calculation during historic and contemporary periods. For the historic period, only a partial proportion of the satellite observations were used. The results demonstrate that the frequency changes calculated from partial observations are closely consistent with those obtained using all data (Supplementary Fig. 12a). We further calculated the standard deviation of the differences between the two across all lakes, representing the uncertainty propagated into the historic frequency due to gaps in remote sensing observations (Supplementary Fig. 12b). This calculated uncertainty was equal to 0.066%. By excluding lakes where the algal bloom frequency less than this uncertainty threshold, we ensured the reliability of our reported frequency change of extremes. This suggests that while observation frequency is an important factor in detecting extreme events, the reported increase in lake extremes observed in our study is not an artifact of differences in observation frequency. This finding underscores the significance of the reported increase in lake extremes and its resilience to potential variations in observation frequency. However, it is important to recognize that observation frequency can influence the detection and characterization of extreme events, and future research may benefit from utilizing higher-frequency observation data to provide a more detailed understanding of extreme event dynamics, which would also aid in improving our understanding of compounding events, where two or more extremes occur sequentially. Nonetheless, our study provides valuable insights into the changing patterns of lake extremes over time, even within the constraints of varying observation frequency.

To explain some of the large-scale variation in algal bloom occurrence in the studied sites, we downloaded information on national scale annual fertilizer consumption from ref. 71. These data describe the estimated total amount of fertilizers per area of cropland per country. In this study, we focus solely on the application of nitrogen (N) and phosphorus (P), measured in kilograms of total nutrient per hectare of cropland, during the study period. For consistency across the studied regions, we used the global dataset provided by ref. 71. to describe large-scale patterns in N and P. However, we note that some countries have other national datasets of N and P which may differ from those provided by Ref. 71. For example, the dataset from National Bureau of Statistics of China (http://www.stats.gov.cn/english/) showed total nutrient and total phosphorus per hectare of cropland were lower than those estimated by ref. 71. (Supplementary Fig. 2). Therefore, we encourage caution in interpreting the large-scale data.

### Identification of increase in univariate extremes and statistical significance
In this study, our central objective was to discern changes in extreme events within lake ecosystems over time. We sought to understand whether there was an increase in the occurrence of these events and, if so, to evaluate the statistical significance of such changes. Our approach involved a comprehensive analysis of two distinct epochs, spanning the periods from the 1980s-1999 and 2010-2019. We examined three key categories of extreme events, namely algal blooms, extreme temperature events, and water level extremes. By comparing the occurrence of these events across the two epochs, we identified instances where a notable increase occurred. This increase was determined by observing a positive change in the frequency of each extreme event. We also employed a bootstrap version (with 1000 iterations) of the Kolmogorov-Smirnov (K-S) test, a widely recognized statistical method for assessing the significance of distributional differences in each extreme event type. The K-S test allowed us to quantitatively evaluate whether the observed changes in each extreme event type (e.g., algal bloom occurrence frequency) was statistically significant between the two epochs or could have arisen due to random variability. The inclusion of the K-S test provided a robust framework for the interpretation of our findings. It enabled us to determine whether the empirical distribution functions of extreme events in the two epochs significantly differed from each other. We also classified the studied lakes as experiencing a notable increase or decrease in the occurrence of univariate, bivariate or multivariate extremes. For extreme temperature events and water level extremes, we determined significance of occurrence changes using statistical test P-values (<0.05) of annual trend changes, leveraging the high-temporal resolution of GLAST and GLEV datasets. Due to limitations of the GBD dataset, we used an alternative method to determine frequency change thresholds corresponding to statistically significant levels. Utilizing the MODIS-derived daily algal bloom dataset for 102 Chinese lakes[9], we computed relative algal bloom frequency changes between historic and present periods, alongside annual trends and P-values. We then grouped relative bloom frequency changes into two categories: significant temporal change (P value ≤ 0.05) and insignificant temporal change (P-value > 0.05). As shown in Supplementary Fig. 13, this approach revealed a distinct threshold value (0.4) effectively distinguishing between the two groups. Despite some overlap, we find this method a viable alternative for quantitatively identifying significant change in bloom frequency.

### Bivariate and multivariate extreme events
Alterations in bivariate extreme events were considered as situations where the frequency of more than one of the univariate extremes described above increased simultaneously within a lake. Specifically, we investigated which lakes experienced parallel changes in the frequency of (i) algal blooms and lake heatwaves, (ii) lake heatwaves and low water extremes, and (iii) algal blooms and low water extremes. We also investigated which lakes experienced an increase in all three extremes during the study period. For each extreme event, we compared their occurrence frequency between the historic (-1980s to 1999) and contemporary (2010 to 2019) period, using all available data. Note that we use a different approach regarding the percentiles used to define extremes for temperature and water level (i.e., lake specific) and algal blooms (i.e., generic across sites). A percentage change in chlorophyll-a, relating to algal blooms, would yield a different result.

## Data availability
The data generated in this study have been deposited in the Figshare database (https://figshare.com/s/d6755addbe4a78a43213). The results generated in this study are also provided in Supplementary Data 1, as well as in the Supplementary Information and as a Source Data file. Source data are provided with this paper.

## Code availability
The code used for the analysis and to produce the figures in this paper are available on Github: https://github.com/gzhaowater/lakeExtremes

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

## Acknowledgements

This work was supported by the UKRI Natural Environment Research Council (NERC): Independent Research Fellowship awarded to RIW [grant number NE/T011246/1]. This work was also supported by NERC grant reference number NE/X019071/1, "UK EO Climate Information Service". This work was also supported by Guangdong Provincial Higher Education Key Technology Innovation Project (2020ZDZX3006). DAD was funded by the Irish HEA Landscape programme and DkIT Research Office. KS was supported by the National Natural Science Foundation of China (U22A20561) and the NIGLAS foundation (E1SL002). GZ was supported by the Chinese Academy of Sciences BR program and the Third Xinjiang Scientific Expedition Program (2021xjkk0803).

## Author contributions

R.I.W. conceived the idea for this study. R.I.W. and G.Z. designed the methodology. G.Z. and Y.T. performed the large-scale computations with input from R.I.W. and L.F. R.I.W., G.Z., Y.T. analyzed the data with input from D.A.D., H.S., L.F., Y.Z., and K.S. R.I.W. led the writing of the manuscript. All authors contributed critically to the drafts and gave final approval for publication.

## Competing interests

The authors declare no competing interests.
