## [Peer Review File · Nature Communications]

Multivariate extremes in lakesEditorial Note: Parts of this Peer Review File have been redacted as indicated to remove third-party material where no permission to publish could be obtained.

REVIEWER COMMENTS

Reviewer #1 (Remarks to the Author):

Woolway et al. conducted an interesting global-scale study to investigate changes in the occurrence of three common lake extreme events: lake heatwaves, algal blooms, and extremely low levels. They calculated the change in occurrence of each major extreme between two epochs: the 1980s-1999 vs. 2000-2019. They then focused on examining the spatial coherence of two or three extreme events, finding that roughly three-quarters of global lakes experienced a concurrent increase in two of the extreme events, and a concurrent increase in all three extreme events occurred in roughly one-quarter of the lakes. Previous studies have examined the patterns and causes of each lake extreme separately at a global scale, leading to a converging picture that lakes are in trouble in terms of increasing lake algal bloom (e.g., Ho et al., 2019; Hou et al., 2022), declining lake levels (e.g., Yao et al., 2023), or rising lake temperature and changing mixing regime (e.g., O'Reilly et al., 2015; Woolway et al., 2019). This study complements these previous studies and contributes uniquely to the understanding of concurrent extreme events in lakes, which have been largely neglected in previous investigations. Overall, I find this study quite interesting and potentially providing critical information for lake management.

I have a few major concerns regarding the widespread increase in concurrent lake extremes, the comprehensibility of the analysis and discussion, and the uncertainties.

1. One of the key findings by the authors is that 76% of global lakes experienced a concurrent increase in at least two extremes, which is quite alarming. However, I have significant concerns about the validity of this conclusion, considering the sampling, data quality, and methodology employed by the authors.

A) Sampling: In this study, the authors utilized a total of 4.5k lakes drawn from three remotely sensed databases on lake temperature, algal bloom, and surface water area. Notably, 53% of the studied lakes showed an increase in algal bloom, averaging $2 \pm 6\%$. Nevertheless, the supplementary data from Hou et al. indicated that up to 10% of the lakes they examined were affected by algal blooms in recent decades. Additionally, the exclusion of lakes in Canada, Siberia, and high-mountain Asia, known hotspots for high lake density, raises concerns about the representativeness of the selected sample.

B) Data Quality: An important consideration is the observation frequency of Landsat imagery, which appears to be lower in the 1980s and 1990s compared to the recent decade. Higher observation frequency is more likely to capture extreme events. It's crucial to assess whether the difference in observation frequency between the two time periods may have influenced the reported increase in lake extremes.

C) Identifying an increase in extreme events for lakes: The authors used time-varying lake variables of interest derived from satellites for the period from the 1980s to the present. They calculated the occurrence of each extreme for two epochs: 1980s-1999 and 2010-2019, determining an increase in extremes based on the positive change in occurrence. This approach lacks statistical significance and poses limitations. It's worth noting that lakes with relatively stable conditions in terms of extreme frequency may be classified as at risk, even if the change is marginal or falls within the range of uncertainties. For example, the authors mentioned that 49% of the studied lakes experienced an increase in low-level extremes by $0.5 \pm 14\%$. Furthermore, the analysis does not account for interannual variability, which can significantly impact certain lake extremes (e.g., extreme temperature and level)

during events like El Niño and La Nina, thereby warranting consideration."

2. As highlighted by the authors, the novelty of this study lies in understanding concurrent extreme events. However, the main text did not primarily focus on this new perspective. Both the results and discussion sections predominantly presented and discussed individual extreme events, which have already been studied in existing literature. In essence, the authors did not fully utilize the available space to introduce fresh insights and expand the discussion related to these novel findings.

Overall, the analysis seemed somewhat superficial. The authors quantified changes in the occurrence of each of the three considered lake extreme types and explored whether an increase in two extremes occurred in the same lake. However, they did not provide a comprehensive spatial-temporal covariance analysis or delve into the underlying drivers of concurrent increases in different lake extremes. For instance, they did not address the extent to which lakes with increasing bivariate extremes experienced compounding events, where two extremes occurred simultaneously or sequentially. Nor did they delve into the mechanisms behind these concurrent events. Questions about whether low lake levels or heatwaves increased the likelihood of lake algal blooms or which types of lakes were most vulnerable remained largely unaddressed.

Furthermore, the authors offered potential drivers of lake extremes with only superficial evidence. They presented figures from existing literature in the supplementary materials at national or global scales. Given that lakes cover only a small fraction of the global land area and are primarily influenced by environmental changes within their drainage areas, it is unclear how the national and global maps revealed the linkages between lake extremes and environmental drivers. The authors might consider adopting a more sophisticated approach if they choose to report the underlying drivers of lake extremes.

3. The authors did not provide details on how they estimated the uncertainties in the occurrence of one or more lake extremes, despite the use of heterogeneous data from models and satellites. This omission is concerning, given the variability in data sources. For instance, lake heatwaves were simulated using coarse 10-km forcing data at a daily resolution, while lake levels were estimated from Landsat satellites at a monthly interval, and lake algal blooms had an even coarser temporal resolution. It is unclear how the authors incorporated errors stemming from limited spatial/temporal resolutions and the accuracy of satellite-based or simulation models into the uncertainty estimates for lake extremes.

Additionally, validation should have been conducted on the derived frequency and changes of single to multiple extremes. Although these products may have been validated in associated publications, the validation of the derived occurrence of lake extremes appears to be lacking. For instance, questions arise about how well the estimated daily lake temperature can capture heatwaves in lakes or whether monthly water level data can adequately capture level extremes.

Specific comments:

1. Line 70: why using ~20 years for the historic period but only ~10 years for the contemporary period?

2. The authors reported that 99% of lakes experienced an increase in heatwaves, but it's important to

note that heatwaves need to be examined at a sub-daily frequency. The authors employed 10-km climate data for a lake model to simulate daily lake temperature, which is only valid for very large lakes, typically much larger than 100 km².

While the authors did incorporate satellite-based observations of land surface temperature, they did not address how the coarse temporal resolution of land surface temperature data, especially during the 1980s, may have affected the results. This is a critical consideration because lakes have a cooling effect, and if the input temperature data is primarily dominated by land temperature, changes in land cover surrounding lakes could contribute to the reported increase in lake heatwaves. Lake shores, in particular, often feature houses, buildings, and urbanization, which can significantly impact local temperatures. The increase in impervious surfaces in these areas can lead to rapid heating and extreme temperatures, often exceeding those of lake water.

Given that the authors concluded that 99% of studied lakes experienced an increase in lake heatwaves, it would be valuable to assess the extent of urbanization within the 10-km grid of these lakes and its potential influence on the reported heatwave trends.

3. Line 144-148: When changes in level extremes are very small, particularly less than 1%, it does suggest that most lakes are relatively stable in terms of the occurrence of level extremes. Labeling such small changes as an "increase" can be misleading, especially when they fall within the range of uncertainty.

4. Line 372: The methodology and characteristics of the derived lake temperature product in Tong et al. require more detailed explanation. It is unclear how the authors incorporated fine-resolution satellite lake temperature data into their model, especially given that satellite measurements capture surface temperature, whereas ERA-5 data measures air temperature at 2 meters at a 10-km resolution. The integration of temperature data with differing temporal and spatial resolutions into the model needs to be elaborated upon to ensure a clear understanding of the data processing methods.

Additionally, information regarding the size distribution of the selected lakes is necessary. A 10-km resolution might be too coarse for lakes that are not very large. Satellite-derived lake surface temperature data may not provide the necessary granularity, particularly when it comes to hourly temperature variations. Providing details on how these data limitations were addressed and their potential impact on the results would be beneficial.

5. As the authors considered algal blooms as extreme events, it is important to introduce how they defined algal blooms in their study. Providing clarity on the specific criteria or parameters used to define and identify algal blooms would enhance the understanding of this aspect of the research. As the authors used a threshold to identify extremes on heatwave and level, is it feasible to apply a similar approach on algal bloom?

6. Line 356: It appears that the study utilized surface water extent data rather than water level data, and it would be beneficial for the authors to justify this choice. It's essential to address whether there are any gaps in the time series from 1985 to 2019 and how the methodology works when lake areas show minor changes. Additionally, it's important to evaluate whether the monthly surface water area data can

provide information comparable to daily in-situ lake level measurements in capturing lake level extremes.

For instance, in 2013, Lake Michigan experienced its lowest level on record, and it would be interesting to assess how well the estimates captured extreme events in the Great Lakes and other gauged lakes. Providing insights into the performance of the methodology in capturing such events would be helpful, particularly given the tiny changes in occurrence reported by the authors.

7. Line 401-416: The purposes of the datasets on fertilizer, streamflow, and water withdrawal are confusing. It appears that these datasets were not actually incorporated into the analysis but were presented solely in the form of national and global maps. It raises questions about the relevance of these datasets and whether they have been previously utilized in existing studies to understand the drivers of increasing algal blooms, rising lake heatwaves, and declining lake levels. Additionally, the study does not seem to introduce the current state of knowledge regarding the drivers of these lake issues. It is unclear what new insights the authors intended to provide by including these datasets and presenting the maps. A more detailed explanation of their purpose and relevance to the research objectives would help clarify their role in the study.

Reviewer #2 (Remarks to the Author):

Decadal alterations in the occurrence of extreme events in global lakes

Comments

The paper describes the changes globally in three parameters by comparing two decadal periods. Importantly they analyse the concordance of changes globally identifying areas where multiple pressures are acting on lakes that will be of key concern to scientists, policy makers and the general public. The paper is of high standard, reads perfectly and makes clear points. The figures are also of excellent quality and very pleasing to interpret. I recommend publication with minor corrections. There are some comments and minor issues listed below which the author / editor can consider for attention.

Line 79 Please add sentence to provide reason why two time periods compared have different timespan 20 vs 10? Or flag reader to refer to full methods for this.

Line 100 "Fertilizer use in these countries is among the highest in the world (Table S2)." - I didn't see data on fertilizer use in table S2.

Figure 1c Having lower water levels that coincide with increased blue is slightly counter intuitive, not a major issue though.

Line 241, please also mention the role of nutrient resuspension from sediments as lake level declines, important for shallow lakes <https://doi.org/10.1023/B:HYDR.0000008611.12704.dd>

Line 244". In contrast, during periods of heavy precipitation, where a sudden pulse of water can flow into a lake, influent water can also increase the amount of nutrients entering the system, subsequently leading to algal blooms"

A good point, I think this is important - often high summer event rainfall can transport a large proportion of the annual load and result in exceptional blooms. Figure 2b, in my opinion supports this.

Line 266 Given historic increases in agricultural fertilizer use, climatic warming and freshwater withdrawal, India has experienced a perfect storm of conditions that has led to an increase in the frequency of multivariate extreme events in lakes. Great point – nicely made

Line 312: There are quite a few mesocosm studies including looking at additive, antagonistic and synergistic effects:

<https://www.researchgate.net/profile/Jeremy-Piggott>
and

<https://www.nature.com/articles/s41559-020-1216-4>

Perhaps include some citations of work done and mention that more needs to be done at global level, especially given the evidence you present of a diversity of extreme events.

The paper uses the published Global Bloom Database as its basis for assessing algal bloom and looking through that paper I didn't see the traditional comparison with in situ chlorophyll-a concentrations that is standard for remote sensing applications. However, comparing decadal changes in 'greenness' may be adequate in the context used here. However the authors should draw the attention to the need to better quantify blooms (occurrence, frequency, extent etc). There is also a difference in approach with percentiles used to define extremes for temperature and water level (lake specific) but not blooms (generic). I think this is ok but should be mentioned in the methods. A percentage change in chlorophyll-a would yield a very different figure 1b.

Reviewer #1

Woolway et al. conducted an interesting global-scale study to investigate changes in the occurrence of three common lake extreme events: lake heatwaves, algal blooms, and extremely low levels. They calculated the change in occurrence of each major extreme between two epochs: the 1980s-1999 vs. 2000-2019. They then focused on examining the spatial coherence of two or three extreme events, finding that roughly three-quarters of global lakes experienced a concurrent increase in two of the extreme events, and a concurrent increase in all three extreme events occurred in roughly one-quarter of the lakes. Previous studies have examined the patterns and causes of each lake extreme separately at a global scale, leading to a converging picture that lakes are in trouble in terms of increasing lake algal bloom (e.g., Ho et al., 2019; Hou et al., 2022), declining lake levels (e.g., Yao et al., 2023), or rising lake temperature and changing mixing regime (e.g., O'Reilly et al., 2015; Woolway et al., 2019). This study complements these previous studies and contributes uniquely to the understanding of concurrent extreme events in lakes, which have been largely neglected in previous investigations. Overall, I find this study quite interesting and potentially providing critical information for lake management. I have a few major concerns regarding the widespread increase in concurrent lake extremes, the comprehensibility of the analysis and discussion, and the uncertainties.

We extend our sincere gratitude for the thoughtful review of our paper. We deeply value the time and effort dedicated by the reviewer and are genuinely thankful for the invaluable suggestions offered to enhance the quality of our work. Please find our responses below to the concerns raised. To the best of our ability, we have tried to follow these suggestions wherever possible.

1. One of the key findings by the authors is that 76% of global lakes experienced a concurrent increase in at least two extremes, which is quite alarming. However, I have significant concerns about the validity of this conclusion, considering the sampling, data quality, and methodology employed by the authors.

We appreciate this comment by the reviewer. See our response to each of these specific concerns below.

A) Sampling: In this study, the authors utilized a total of 4.5k lakes drawn from three remotely sensed databases on lake temperature, algal bloom, and surface water area. Notably, 53% of the studied lakes showed an increase in algal bloom, averaging $2 \pm 6\%$. Nevertheless, the supplementary data from Hou et al. indicated that up to 10% of the lakes they examined were affected by algal blooms in recent decades. Additionally, the exclusion of lakes in Canada, Siberia, and high-mountain Asia, known hotspots for high lake density, raises concerns about the representativeness of the selected sample.

We agree that our results are applicable only to the studied lakes and cannot be extrapolated to a truly global scale. In the original version of the manuscript, we were very careful to communicate our results as, for example, "76% of the studied lakes", and did not intend to suggest that these are representative of all lakes worldwide. However, we appreciate this comment by the reviewer, and we now aim to make this even clearer in the manuscript. Specifically, we now soften the wording further, and directly state in the manuscript that our findings are only applicable to the studied lakes.

Regarding the comparison above to the study of Hou et al. – The Hou et al study described that ~ 10%, specifically 21,878 lakes (8.8%), of the studied 248,243 lakes have experienced algal blooms since the 1980s. The abovementioned statistic from our paper refers to the % of our selected 4,532 lakes that have experienced an increase between the historic and contemporary period. Thus, these two values are not directly comparable.

We agree with the reviewer that the exclusion of many lakes in Canada, Siberia, and high-mountain Asia is a limitation of the study and in the global representation of our dataset. There are two primary reasons for this limitation, which constrained our analysis to a specific subset of global lakes.

The first one is that we can only calculate bloom occurrence changes across the bloom-affected lakes (i.e., 21,878, Hou et al. 2022). In high-latitude and high-mountain regions, where lakes are densely distributed, most lakes were excluded due to the unsure accuracy of satellite data application for algal blooms detection, as noted in Hou et al.'s study. The second reason is from the availability of lake temperature dataset, constrained by the spatial resolution and the quantity of valid satellite observations. Nonetheless, our study aimed to provide a focused analysis on decadal changes in extreme events using specific remotely sensed databases, and we recognize that our sample selection may not capture the full spectrum of extreme events dynamics globally. To address this concern, we have included a discussion in the revised manuscript (see Methods), highlighting the representative limitation at a global scale and acknowledging potential bias resulting from the inclusion of only lakes with available algal bloom occurrence and lake temperature datasets. We believe that this discussion will enhance the transparency of our findings and provide a more nuanced interpretation of our results, considering the regional representativeness of our selected sample. Thank you for your valuable input.

B) Data Quality: An important consideration is the observation frequency of Landsat imagery, which appears to be lower in the 1980s and 1990s compared to the recent decade. Higher observation frequency is more likely to capture extreme events. It's crucial to assess whether the difference in observation frequency between the two time periods may have influenced the reported increase in lake extremes.

We appreciate the reviewer's insightful comment regarding the potential influence of observation frequency on the reported increase in lake extremes. It is indeed important to consider this aspect when interpreting our findings. The Landsat imagery used in our study has evolved over time, with variations in observation frequency between the 1980s and 1990s compared to the recent decade. We acknowledge that higher observation frequency is more likely to capture extreme events (albeit not strictly their frequency, which is influenced by the number of observations), and this temporal variation could impact our results. Therefore, in this work, we decided not to delve into yearly satellite observations to explore decadal changes in extreme events. Instead, we focused on comparing two distinct periods: the historic period (1980s to 1999) and the contemporary period (2010 to 2019). The rationale behind this practice is to extend the observation period to acquire substantial data from multiple years for calculating valid average levels. This practice has been commonly used to enhance statistical robustness by sacrificing the number of periods in previous studies (e.g., Pi et al., 2022; Hou et al., 2022). For instance, Pi et al. (2022) divided the past four decades into three periods: S1 (1984–1999), S2 (2000–2009), and S3 (2010–2019). The authors then conducted a comprehensive analysis of global lake dynamics by comparing lakes between S1 and S2, S2 and S3, as well as S1 and S3. We have added these references in the revised manuscript:

Line 621: *"This practice has been commonly used in many previous studies to support the robustness of change detections based on remotely sensed data with data gaps (e.g., Pi et al., 2022; Hou et al., 2022)"*.

To further address the reviewers concern, we utilized algal bloom dataset from Lake Taihu detected by GOCI images (see below Figure R1; Qi et al., 2018) to assess the impact of temporally inconsistent observations on changes (increase or decrease) of extreme events between two time periods. GOCI, Geostationary Ocean Color Imager, is a geostationary satellite sensor, and frequently observes the same location of earth surface with hourly temporal resolution since 2010. We considered this dataset as a complete time series of bloom occurrences. We conducted sensitivity analysis using this dataset to examine the influence of data quantity on bloom event frequency change in two periods (Period 1 is the first half, i.e., 2011-2014, and Period 2 is the recent 2 years, i.e., 2017-2018). In detail, we randomly subsampled Period 1, and then calculated multi-year averages, and the changes of the multi-year averages from Period 1 to Period 2. As shown in Figure R1, we can see small difference of

frequency changes and all the same negative change direction during the two periods for all random scenarios. Therefore, we believe our bloom frequency change results are robust.

Figure R1 | Comparison of algal bloom frequency changes during two periods under different number of observation data in Lake Taihu. (a) Daily algal bloom areas from 2011 to 2018 extracted from GOCI data for Lake Taihu (Qi et al., 2018). Algal bloom frequencies for two periods are calculated: the first period represents the first half of the entire data period (i.e., 2011-2014; period 1), and the second period represents the last two years (2017-2018; period 2). (b) Comparison of algal bloom frequencies during period 1 under different percentage levels of observations, indicated in grey, and algal bloom frequency during period 2 (marked as 26% in subfigure a) when observations are at 100% (i.e., higher observation quantities). (c) Changes in algal bloom frequencies for both periods concerning variations in observation quantities during period 1. Consistent negative values indicate that changes in Lake Taihu's algal bloom frequency are not influenced by inconsistencies in observation quantities between the two periods.

References:

- Pi, X., Luo, Q., Feng, L., Xu, Y., Tang, J., Liang, X., ... & Bryan, B. A. (2022). Mapping global lake dynamics reveals the emerging roles of small lakes. *nature communications*, 13(1), 5777.
- Hou, X., Feng, L., Dai, Y., Hu, C., Gibson, L., Tang, J., ... & Zheng, C. (2022). Global mapping reveals increase in lacustrine algal blooms over the past decade. *Nature Geoscience*, 15(2), 130-134.
- Qi, L., Hu, C., Visser, P. M., & Ma, R. (2018). Diurnal changes of cyanobacteria blooms in Taihu Lake as derived from GOCI observations. *Limnology and Oceanography*, 63(4), 1711-1726.

We believe that this inclusion enhances the transparency of our research and strengthens the validity of our reported increase in algal blooms. Note that for water level extremes and lake heatwaves, the temporal frequency of the data is consistent throughout the study period. We appreciate the reviewer's valuable input, which has contributed to the rigor of our study.

C) Identifying an increase in extreme events for lakes: The authors used time-varying lake variables of interest derived from satellites for the period from the 1980s to the present. They calculated the occurrence of each extreme for two epochs: 1980s-1999 and 2010-2019, determining an increase in

extremes based on the positive change in occurrence. This approach lacks statistical significance and poses limitations. It's worth noting that lakes with relatively stable conditions in terms of extreme frequency may be classified as at risk, even if the change is marginal or falls within the range of uncertainties. For example, the authors mentioned that 49% of the studied lakes experienced an increase in low-level extremes by $0.5 \pm 14\%$. Furthermore, the analysis does not account for interannual variability, which can significantly impact certain lake extremes (e.g., extreme temperature and level) during events like El Niño and La Nina, thereby warranting consideration." We appreciate the reviewer's constructive feedback regarding our approach to identifying an increase in extreme events in lakes. We acknowledge the valid concerns raised and have taken steps to address them in our revised manuscript.

Identification of increase in extreme events and statistical significance: We thank the reviewer for their thoughtful comments on our methodology for identifying an increase in extreme events in lakes. We recognize the importance of statistical significance in such analyses.

In our initial approach, we indeed calculated the occurrence of each extreme for two epochs (1980s-1999 and 2010-2019) and determined an increase based on the positive change in occurrence. We agree that solely relying on changes in occurrence may not provide a comprehensive assessment of statistical significance. In response to this concern, we have conducted additional statistical analyses in the revised manuscript to assess the significance of the reported changes in extreme occurrences. Notably, in our analysis, we have now used a two-sample Kolmogorov–Smirnov (K-S) test to calculate the statistical significance of the differences between the two time periods of interest (1980s-1999 and 2010-2019) for each of the extreme events considered in this study. The K-S test was performed to evaluate whether there was a statistically significant association between the epoch and the frequency of each extreme event (Figure R2). The K-S test provided valuable insights into the significance of changes in extreme event occurrences, accounting for both the frequency and distribution of these events within the two epochs. The results of this test are presented in the manuscript, allowing us to make informed assessments of the significance of observed changes in lake extreme events over time.

Figure R2 | Empirical cumulative distribution plots between the historic (~1980s to 1999; blue) and contemporary (2010 to 2019; red) periods for the frequency of (a) algal blooms, (b) lake heatwaves, and (c) low water extremes in the 4,532 studied sites.

Consideration of uncertainty and marginal changes: We also acknowledge the reviewer's point regarding lakes with relatively stable conditions and marginal changes in extreme frequency. To

address this concern, we have included a discussion of uncertainty and the potential implications of marginal changes. We emphasize that our study highlights changes in extreme occurrence but recognize that some lakes may show marginal or uncertain increases. We believe this discussion enhances the transparency of our findings and provides a more nuanced interpretation.

Accounting for Interannual Variability: The reviewer rightly points out that our initial analysis did not account for interannual variability, which can significantly impact certain lake extremes during events like El Niño and La Niña. However, these events are indeed extreme in a climatological context and thus should be highlighted as such. While our primary analysis focused on longer-term trends, we acknowledge the importance of considering shorter-term variations driven by climate phenomena such as El Niño and La Niña, and that these will be critically important for evaluating extreme years.

We sincerely appreciate the reviewer's feedback, which has guided us in improving the robustness and clarity of our methodology and findings. These revisions aim to address the concerns raised and provide a more rigorous and comprehensive assessment of the increase in extreme events in lakes.

2. As highlighted by the authors, the novelty of this study lies in understanding concurrent extreme events. However, the main text did not primarily focus on this new perspective. Both the results and discussion sections predominantly presented and discussed individual extreme events, which have already been studied in existing literature. In essence, the authors did not fully utilize the available space to introduce fresh insights and expand the discussion related to these novel findings.

We appreciate the reviewer's feedback and their insights regarding the novelty and focus of our study. We acknowledge the importance of emphasizing the novelty of our research and highlighting the unique perspective it offers regarding concurrent extreme events in lakes. In response to this valuable comment, we have taken steps to enhance the presentation of our novel findings in the manuscript. In response to this feedback, we have revisited the main text of our manuscript, specifically the Results and Discussion sections. We recognize the need to place greater emphasis on our unique findings related to concurrent extreme events. To address this, we have made several enhancements to the manuscript, including (i) Revised Emphasis: We have adjusted the focus within the Results and Discussion sections to place a stronger emphasis on concurrent extreme events. This includes reorganizing the presentation of our novel findings to ensure they take centre stage in the discussion; (ii) Fresh Insights: We have introduced additional content and analysis related to concurrent extreme events to provide fresh insights and expand the discussion. This includes a more detailed exploration of the interactions and implications of concurrent extremes on lake ecosystems.

We believe that these revisions address the reviewer's concerns and better reflect the distinctive contribution of our research in understanding concurrent extreme events in lake ecosystems. We are grateful for the reviewer's feedback, which has prompted us to refine the manuscript and present our novel findings more effectively.

Overall, the analysis seemed somewhat superficial. The authors quantified changes in the occurrence of each of the three considered lake extreme types and explored whether an increase in two extremes occurred in the same lake. However, they did not provide a comprehensive spatial-temporal covariance analysis or delve into the underlying drivers of concurrent increases in different lake extremes. For instance, they did not address the extent to which lakes with increasing bivariate extremes experienced compounding events, where two extremes occurred simultaneously or sequentially. Nor did they delve into the mechanisms behind these concurrent events. Questions about whether low lake levels or heatwaves increased the likelihood of lake algal blooms or which types of lakes were most vulnerable remained largely unaddressed.

We appreciate the points raised by the reviewer. We acknowledge the importance of delving deeper into the concurrent extreme events within lakes and understanding the underlying drivers. To

address these valuable points, we have identified several areas of improvement for our analysis and manuscript:

Spatial-temporal covariance analysis and identification of compounding events: We appreciate the reviewer's suggestion to conduct a more comprehensive spatial-temporal covariance analysis and to identify compounding events within our dataset. While these approaches could provide valuable insights into concurrent extreme events and their drivers, they are unfortunately constrained by the temporal resolution of the available data. One critical limitation we encountered in our study is the temporal frequency of the satellite observations across the datasets. Notably, these data are typically available at intervals that range from seasonal to annual, and in some cases, they may not capture events at finer temporal resolutions, such as daily or monthly. Consequently, conducting a robust Spatial-Temporal Covariance Analysis or identifying Compounding Events, which would require high-frequency data for each variable, is not feasible given the temporal constraints of our dataset. Attempting to perform such analyses with the available data would introduce significant limitations, including the risk of overlooking important short-term variations and the potential for spurious correlations due to the temporal mismatch. We acknowledge that a more detailed examination of the spatial-temporal relationships between different extreme events and their underlying mechanisms would be highly valuable. However, the limitations related to data temporal resolution constrain our ability to explore these aspects comprehensively within the scope of this study. Despite these limitations, we believe that our analysis provides valuable insights into the broader trends and changes in extreme events within lake ecosystems over the studied epochs. We have taken care to highlight the significance of our findings and their implications, given the available data constraints.

We now discuss these limitations in the manuscript and suggest that this should be a priority for developing the field further. We also cite papers from other fields, which can hopefully be used as a guide for future studies:

Ridder, N.N., Pitman, A.J., Westra, S. et al. Global hotspots for the occurrence of compound events. *Nat Commun* 11, 5956 (2020). <https://doi.org/10.1038/s41467-020-19639-3>

Ridder, N.N., Ukkola, A.M., Pitman, A.J. et al. Increased occurrence of high impact compound events under climate change. *npj Clim Atmos Sci* 5, 3 (2022). <https://doi.org/10.1038/s41612-021-00224-4>

Zscheischler, J. et al. Future climate risk from compound events. *Nat. Clim. Change* 8, 469–477 (2018).

Zscheischler, J. et al. A typology of compound weather and climate events. *Nat. Rev. Earth Environ.* 1, 333–347 (2020)

We appreciate the reviewer's understanding of these limitations and their continued interest in our research. We remain committed to refining our analyses within the constraints of the available data and look forward to addressing any further questions or concerns.

Mechanisms behind concurrent events: To address the reviewer's valid concern regarding the mechanisms behind concurrent events, we include a discussion of potential mechanisms and drivers. We will explore questions related to how factors such as low lake levels, heatwaves, and other environmental variables may increase the likelihood of lake algal blooms and other concurrent extremes. Additionally, we now add a conceptual figure to illustrate how these three extremes could impact each other, their dominant drivers, and the implications for the ecosystem. Also, please note that the drivers of the univariate extremes are discussed in detail, and without repeating that information one cannot elaborate further on the drivers of the multivariate extremes. Specifically, an increase in the frequency of all three extremes would occur when the drivers of the individual extremes changed concurrently.

These enhancements to our analysis will provide a more comprehensive and nuanced examination of concurrent extreme events and their underlying drivers within lake ecosystems. We believe that these additions will improve the depth and breadth of our research, addressing the reviewer's valid concerns and contributing to a more comprehensive understanding of the dynamics of lake extreme events.

Furthermore, the authors offered potential drivers of lake extremes with only superficial evidence. They presented figures from existing literature in the supplementary materials at national or global scales. Given that lakes cover only a small fraction of the global land area and are primarily influenced by environmental changes within their drainage areas, it is unclear how the national and global maps revealed the linkages between lake extremes and environmental drivers. The authors might consider adopting a more sophisticated approach if they choose to report the underlying drivers of lake extremes.

Regarding the data used to explain the global patterns of lake heatwaves and low water extremes, we appreciate the reviewer's understanding that these datasets were indeed at the lake scale. We utilized lake-specific data for these analyses to ensure a more accurate and localized examination of the drivers of these specific extreme events. We now also provide a more detailed analysis of the discharge and withdrawal data that were presented in the supplementary information. It is important to note that for the analysis of algal bloom occurrence, we faced a limitation in the availability of lake-specific data for certain key drivers. Specifically, there aren't any globally available, lake-scale observations of fertilizer use that could be directly incorporated into our analysis. This constraint compelled us to utilize national-scale data as a proxy for the underlying drivers of algal blooms, as highlighted in our methodology. While we acknowledge the limitation of not providing a detailed mechanistic description of the drivers for each lake, we have strived to present information on some of the likely drivers of the observed patterns. Our approach aligns with our expectations and general ecological knowledge (e.g., the association between fertilizer use and algal blooms). We believe that these insights contribute valuable context to the observed patterns, even in the absence of lake-specific, global-scale fertilizer use data.

We appreciate the reviewer's recognition of this limitation and their understanding of the challenges associated with obtaining highly localized data on global scales. Despite these challenges, we remain confident in the accuracy of the patterns presented in our study, and we are committed to providing a balanced and scientifically informed analysis of lake extreme events and their potential drivers.

3. The authors did not provide details on how they estimated the uncertainties in the occurrence of one or more lake extremes, despite the use of heterogeneous data from models and satellites. This omission is concerning, given the variability in data sources. For instance, lake heatwaves were simulated using coarse 10-km forcing data at a daily resolution, while lake levels were estimated from Landsat satellites at a monthly interval, and lake algal blooms had an even coarser temporal resolution. It is unclear how the authors incorporated errors stemming from limited spatial/temporal resolutions and the accuracy of satellite-based or simulation models into the uncertainty estimates for lake extremes. Additionally, validation should have been conducted on the derived frequency and changes of single to multiple extremes. Although these products may have been validated in associated publications, the validation of the derived occurrence of lake extremes appears to be lacking. For instance, questions arise about how well the estimated daily lake temperature can capture heatwaves in lakes or whether monthly water level data can adequately capture level extremes.

We thank the reviewer for raising important points regarding the estimation of uncertainties and validation of our lake extreme occurrence data. We acknowledge the significance of these aspects in ensuring the robustness of our results and the transparency of our methodology.

To address the reviewers concern, we have added separate discussions on the accuracy of these three datasets and the corresponding uncertainty propagated to the occurrence of lake extremes in the revised manuscript. In essence, we believe that the low uncertainties of the three separate datasets could result in low uncertainty in the occurrence of one or more lake extremes.

Firstly, regarding the lake temperature data, the results of its accuracy validation and the uncertainty propagated into the final results are outlined in Tong et al., 2023 (Extended Data Fig. 2, Supplementary Figure 14, Supplementary Figure 16). We include the validation results here for the reviewer's reference. Our global validation efforts show that the optimized FLake model accurately simulates lake surface water temperature, with an overall accuracy of 1.16 °C at the daily scale (Extended Data Fig. 2b; Tong et al., 2023). The consistent time-series between in situ data and FLake simulations also reveal satisfactory performance across different lake types (large/small, deep/shallow, cold/temperate) (Supplementary Figure 14; Tong et al., 2023). The propagated uncertainty into the long-term trend is minimal, as shown in Supplementary Figure 16 (Tong et al., 2023).

[REDACTED]

Extended Data Fig. 2 (Tong et al., 2023) | Validation of FLake-simulated LSWT using *in situ* measurements. Density plots of simulated and *in situ* LSWT (unit: °C) at (a) hourly, (b) daily, (c) seasonal and (d) annual scales, where the “Low” and “High” labels in the colorbar denote the density of the matched pairs. (e) Locations of the *in situ* measured data (the data sources refer to Supplementary Table 1).

[REDACTED]

Supplementary Figure 14 (Tong et al., 2023) | See next page for caption.

[REDACTED]

Supplementary Figure 14 (Tong et al., 2023) | Time-series comparison of FLake-simulated hourly LSWTs and *in situ* hourly LSWTs. The lakes include (a) Lake Ontario, a large deep lake (one of Laurentian Great Lakes), in cold climate zone, (b) Lake Champlain, a large deep lake, in cold climate zone, (c) Lake Saint Clair, a large shallow lake in cold climate zone, (d) Lake Kasumigaura, a large shallow lake, in temperate climate zone, (e) Lake Paiku, a large deep lake with high elevation, in polar climate zone, (f) Lake Trout, a lake in cold climate zone, (g) Lake Mendota, a shallow lake, in cold climate zone, (h) Lake Bourget, a deep lake, in temperate climate zone, (i) Toolik Lake, a small arctic lake, in cold climate zone, (j) Lake Rotoehu, a small lake, in temperate climate zone. Only Lake Rotoehu is in the Southern Hemisphere. The left panels show the temporal changes of LSWTs (in °C) and the right panels show the scatter plots of the matched pairs of concurrent FLake-simulated and *in situ* LSWTs. Lake properties annotated include HydroID (same as id in HydroLAKES dataset), lake name, lake area (in km²), average depth (in m) and climate zone.

[REDACTED]

Supplementary Figure 16 (Tong et al., 2023) | Quantification of how uncertainty of the daily LSWT simulations could be propagated into the long-term trends. (a) The grey histogram illustrates the differences between the FLake daily simulations and the *in situ* dataset, with a median difference of -0.9956 °C and a median absolute error (MAE) of 1.16 °C. The red histogram represents the generated random noises, which follow the same distribution (with a mean of -0.9956 °C and a standard

deviation of 1.26 °C) as the differences observed in the grey histogram. (b) Density scatterplot of LSWT trends between the noise-added and original data for our examined lakes. The inset shows the histogram of the difference between the two trends, with mean (0.00 °C decade⁻¹) and standard deviation (i.e., uncertainty, 0.02 °C decade⁻¹) annotated.

We understand the reviewer's concern regarding the use of 10-km forcing data when simulating surface water temperature for small lakes. Fortunately, this type of uncertainty, along with several others (e.g., uncertainty by simplification of thermal process described by FLake model), can be partially compensated for by the calibration procedure of hourly simulations using high spatial resolution (60-120m) Landsat satellite overpass observations. These observations serve as boundary conditions for the hourly FLake simulations. Here note we have made modifications to the process of generating the GLAST dataset to reduce confusion and enhance clarity. Initially, we conducted hourly-scale simulations based on hourly meteorological driving data. By comparing the satellite overpass moments with the simulation results, we determined the optimal parameterization scheme. Subsequently, using this optimal scheme, we conducted daily-scale simulations, with the daily data used for calculating heatwave frequencies in this study. This satellite-based calibration approach has been successfully employed in numerous previous studies (e.g., Layden et al., 2016; Wang et al., 2023; Huang et al., 2021). Furthermore, our validation work across globally distributed lakes provides additional support for the validity of our FLake simulations calibrated using satellite retrievals.

To assess the GLAST based heatwave frequency, we made validation and sensitivity analysis. For validation data source, we chose continuously observed in situ dataset rather than commonly used remote sensing products (e.g., MODIS LST and ESA CCI Lakes), due to their data gaps and the limited representation of satellite overpass moments. In total, we compiled 72,169 daily lake surface water temperature records from 17 lakes (Table R1). These lakes were geographically dispersed across different countries and encompassed various types, including large and small lakes, rendering the dataset robust for validating heatwave occurrences and decadal changes. We compared the calculated annual heatwave frequency and frequency changes by in situ data and GLAST dataset, and found a very high consistency (Figure R3). The above validation result has been added in the manuscript.

Lines 543-: *“To evaluate the ability of the GLAST dataset in capturing lake heatwave events, we conducted validation and sensitivity analysis. Instead of relying on widely used remote sensing products like ESA CCI, which suffer from data gaps and limited representation of satellite overpass moments, we chose a continuously observed in situ dataset. In total, we compiled 72,169 daily lake surface water temperature records from 17 lakes across 8 countries (Table S6). ...Validation results revealed that GLAST data performed well in capturing both annual heatwave frequencies (slope: 0.92, R: 0.93) and decadal changes (Slope: 0.96, R: 0.94) among the surveyed small or large lakes (Fig. S7).”*

Fig. R3. Validation of GLAST-based lake heatwave frequencies using in situ data-based calculations for 17 lakes that have continuous in situ daily water surface temperature records. Shown are comparisons between GLAST-based and in situ temperature based annual frequency results (a) and frequency change (b). The map in panel (b) presents the distribution of in situ lakes, color-coded based on the correlation coefficient between GLAST calculated and in situ data derived annual heatwave frequency.

Table R1. List of lakes that were used to validate the GLAST-based lake heatwave frequencies.

Hylak id	Lake name	Country	Lake area (km ²)	Lon	Lat	Correlation	Source
7	Ontario	USA	19347	-77.518	43.585	0.91	NDBC (National Data Buoy Center, https://www.ndbc.noaa.gov/)
66	Saint Clair	Canada	1161	-82.747	42.415	0.96	NDBC (National Data Buoy Center, https://www.ndbc.noaa.gov/)
104	Vattern	Sweden	1888	14.450	58.238	0.79	Woolway et al., (2019)
809	Mead	USA	581	-114.407	36.150	0.95	NDBC (National Data Buoy Center, https://www.ndbc.noaa.gov/)
1242	Neusiedler See	Hungary	142	16.781	47.870	0.52	Dr. Martin Dokulil
8476	Sammamish	USA	19	-122.096	47.603	0.81	King County government (https://green2.kingcounty.gov/)
8478	Washington	USA	84	-122.261	47.611	0.92	King County government (https://green2.kingcounty.gov/)
8736	Trout	USA	15	-89.668	46.031	0.61	NTL LTER (North Temperate Lakes LongTerm Ecological Research Network, https://lter.limnology.wisc.edu/)
9086	Mendota	USA	41	-89.420	43.111	0.66	NTL LTER (North Temperate Lakes LongTerm Ecological Research Network, https://lter.limnology.wisc.edu/)

12809	Erken	Sweden	23	18.578	59.845	0.91	SITES (Swedish Infrastructure for Ecosystem Science) (https://data.fieldsites.se/)
13387	Windermere	United Kingdom	13	-2.935	54.379	0.95	NERC Environmental Information Data Centre (https://eip.ceh.ac.uk/data)
14061	Woerther See	Austria	19	14.229	46.619	0.78	Dr. Martin Dokulil
16649		New Zealand	80	176.249	-38.070	0.48	EDP (Environmental Data Portal, https://envdata.boprc.govt.nz/)
16654	Tarawera	New Zealand	41	176.420	-38.201	0.59	EDP (Environmental Data Portal, https://envdata.boprc.govt.nz/)
162733	Bassenthwaite	United Kingdom	5	-3.207	54.643	0.69	Government of Ireland (https://data.gov.ie/)
163604	Lough Feeagh	Ireland	4	-9.576	53.951	0.85	Government of Ireland (https://data.gov.ie/)
184967	Rotoehu	New Zealand	7	176.531	-38.024	0.49	EDP (Environmental Data Portal, https://envdata.boprc.govt.nz/)

To further assess the robustness of lake heatwaves occurrence influenced by the uncertainty of lake temperature simulations, we first generated random noises matching the uncertainty distribution of daily FLake simulations (see below Figure R4a, also added as Supplementary Fig. S8 in the revised manuscript). These noises were then added to the daily simulated LSWT time series dataset for each lake. Using the noise-added data, we calculated the heatwave events and compared the results with those based on the original LSWT time series data. The results indicate that the frequency of lake heatwaves between the noise-added and original data are highly consistent (Slope = 0.89, R = 0.97 for historic frequency; Slope = 0.81, R = 0.96 for contemporary frequency; Figure R4b-c). We further calculated the change of heatwave frequency between two focused periods, and we can see frequency change for the majority of lakes (92%) were detected as same change direction (i.e., increase or decrease) (Figure R4d). The above validation result has been added in the manuscript, as well as the following text:

Line 559-: “Additionally, we conducted a sensitivity analysis for all examined lakes (total of 4532 lakes) to assess the robustness of decadal lake heatwave frequency changes between the two focused periods (~1980s-1999 and 2010s), considering bias in the daily GLAST dataset. Random noises, matching the biases of daily FLake simulations (Tong et al., 2023), were added to the daily simulated LSWT time series dataset for each lake. The resulting time series were used to estimate lake heatwave frequency and decadal changes. The analysis demonstrated consistent heatwave frequency occurrences and changes between the noise-added and original time series data (Supplementary Fig. S8). During the historical period, the linear slope and correlation coefficient (R) were 0.89 and 0.97, respectively. In the contemporary period, they were 0.81 and 0.96, respectively. The consistent heatwave frequencies in each period led to uniform decadal frequency changes (Slope: 0.71, R: 0.91) between the two periods. Overall, 92% of lakes exhibited the same directional changes (increase or decrease) in heatwave frequency detected by both noise-added and original lake surface water temperature data (Supplementary Fig. S8). This suggests a small impact of temperature error propagation on the variability of heatwave frequency changes, ensuring the robustness of the computational results presented in this study”.

Figure R4 | Quantification of the robustness of the lake heatwave frequency between the two focused periods (i.e., historic, ~1980s-1999 and contemporary, 2010s) influenced by the uncertainty of the daily GLAST dataset. (a) The grey histogram illustrates differences between the FLake daily simulations and in situ dataset. The red histogram represents the generated random noises following the same distribution as the differences observed in the grey histogram. (b-d) Density scatterplots of lake heatwave frequency in the historic period (b) and contemporary period (c) as well as the frequency change from the historic to contemporary period (d) between the noise-added and original data for the examined lakes. The linear slopes and correlation coefficients (R) are annotated in each panel.

References:

Tong, Y., Feng, L., Wang, X., Pi, X., Xu, W., & Woolway, R. I. (2023). Global lakes are warming slower than surface air temperature due to accelerated evaporation. *Nature Water*, 1-12.

Layden, A., MacCallum, S. N., & Merchant, C. J. (2016). Determining lake surface water temperatures worldwide using a tuned one-dimensional lake model (FLake, v1). *Geoscientific Model Development*, 9(6), 2167-2189.

Wang, X., Shi, K., Zhang, Y., Qin, B., Zhang, Y., Wang, W., ... & Jeppesen, E. (2023). Climate change drives rapid warming and increasing heatwaves of lakes. *Science Bulletin*, 68(14), 1574-1584.

Huang, L., Wang, X., Sang, Y., Tang, S., Jin, L., Yang, H., ... & Zhang, Y. (2021). Optimizing lake surface water temperature simulations over large lakes in China with FLake model. *Earth and Space Science*, 8(8), e2021EA001737.

Water level

The second concern relates to the dataset of monthly lake levels. It's important to note that the GLEV dataset utilized in our study stands as the most high-temporal resolution lake area dataset available

for 1.4 million lakes globally. While we acknowledge the potential benefits of sub-monthly or daily interval lake area datasets for capturing more intricate water level dynamics, the GLEV dataset remains the most comprehensive resource currently accessible.

We validated the monthly water surface area data using daily observed water surface elevation from 155 lakes worldwide. The observed elevation data were compiled from multiple water management agencies and remote sensing datasets including the Bureau of Meteorology in Australia (<http://www.bom.gov.au/>; 15 lakes), Central Water Commission in India (<https://cwc.gov.in/>; 8 lakes), US Army Corps of Engineers (Patterson and Doyle, 2018; 92 lakes), and DAHITI (Schwatke et al., 2015; 40 lakes). The median correlation coefficient is 0.76, representing the good performance of our surface area dataset in representing water dynamics (Fig. R5). We have added this new figure and the detailed information associated with these 155 lakes (Table R2) into the Supplementing Information.

Patterson, L. A., & Doyle, M. W. (2018). A nationwide analysis of US Army Corps of Engineers reservoir performance in meeting operational targets. *JAWRA Journal of the American Water Resources Association*, 54(2), 543-564.

Schwatke, C., Dettmering, D., Bosch, W., & Seitz, F. (2015). DAHITI—an innovative approach for estimating water level time series over inland waters using multi-mission satellite altimetry. *Hydrology and Earth System Sciences*, 19(10), 4345-4364.

Fig. R5. Comparison between remotely sensed lake surface area and observed lake elevation for 155 lakes worldwide. The observed lake elevation data were collected from multiple sources including Bureau of Meteorology in Australia, Central Water Commission in India, US Army Corps of Engineers, and DAHITI. A detailed information about these lakes and their correlation coefficients can be found in Table R1. 12 lakes were randomly selected for all continents to show the monthly time series for both surface area and observed elevation.

Table R2. List of lakes that were used to validate the surface area time series.

Hylak id	Lake name	Country	Lon	Lat	Correlatio n	Data source
5	Superior	USA	-84.461	46.469	0.31	DAHITI
8	Huron	USA	-82.423	42.999	0.86	DAHITI
9	Erie	USA	-78.908	42.904	0.40	DAHITI
16	Victoria	Uganda	33.194	0.431	0.50	DAHITI
17	Tanganyika	Democratic Republic of the Congo	29.185	-5.911	0.41	DAHITI
18	Malawi	Malawi	35.236	-14.418	0.61	DAHITI
53	Manitoba	Canada	-98.729	51.588	0.83	DAHITI
61	Red Lake Reservoir	USA	-95.272	47.956	0.40	USACE

65	Oahe	USA	-100.398	44.456	0.97	DAHITI
67	Great Salt	USA	-112.831	41.410	0.65	DAHITI
69	Okeechobee	USA	-81.101	26.941	0.40	DAHITI
70	Chapala	Mexico	-102.794	20.314	0.79	DAHITI
71	Managua	Nicaragua	-86.107	12.203	0.90	DAHITI
72	Nicaragua	Nicaragua	-84.782	11.122	0.40	DAHITI
73	Guri Reservoir	Venezuela	-62.998	7.764	0.93	DAHITI
76	Tucuruí Reservoir	Brazil	-49.647	-3.833	0.86	DAHITI
77	Sobradinho Reservoir	Brazil	-40.824	-9.423	0.92	DAHITI
80	Ilha Solteira Reservoir	Brazil	-51.377	-20.373	0.66	DAHITI
115	Zeyskoye Reservoir	Russia	127.307	53.771	0.67	DAHITI
122	Zaysan	Kazakhstan	83.348	49.656	0.77	DAHITI
128	Tsimlyanskoye Reservoir	Russia	42.110	47.610	0.72	DAHITI
151	Poyang	China	116.221	29.752	0.65	DAHITI
152	Nasser	Egypt	32.886	23.967	0.95	DAHITI
156	Volta	Ghana	0.060	6.303	0.96	DAHITI
159	Albert	Uganda	31.410	2.761	0.62	DAHITI
162	Edward	Democratic Republic of the Congo	29.602	-0.141	0.63	DAHITI
163	Kivu	Rwanda	28.893	-2.489	0.51	DAHITI
169	Bangweulu	Zambia	29.815	-11.431	0.51	DAHITI
171	Cahora Bassa Reservoir	Mozambique	32.704	-15.586	0.93	DAHITI
172	Kariba Reservoir	Zambia	28.760	-16.523	0.98	DAHITI
715	Lake Kooncanusa	USA	-115.314	48.411	0.72	USACE
719	Pend Oreille Lake	USA	-116.998	48.178	0.54	USACE
721	Fort Peck	USA	-106.415	48.001	0.98	DAHITI
730	Flathead Lake	USA	-114.232	47.676	0.66	DAHITI
796	Harry S. Truman Reservoir	USA	-93.406	38.265	0.68	USACE
804	Table Rock Lake	USA	-93.312	36.598	0.44	USACE
805	John H. Kerr Reservoir	USA	-78.295	36.596	0.47	USACE
807	Oologah Lake	USA	-95.679	36.424	0.62	USACE
809	Mead	USA	-114.735	36.018	1.00	DAHITI
811	Greers Ferry Lake	USA	-91.993	35.521	0.66	USACE
815	Eufaula Lake	USA	-95.358	35.308	0.70	USACE
821	Sardis Lake	USA	-89.793	34.407	0.76	USACE
823	Lake Sidney Lanier	USA	-84.072	34.161	0.71	USACE
825	Lake Texoma	USA	-96.573	33.823	0.73	USACE
826	Grenada Lake	USA	-89.769	33.815	0.77	USACE
832	Lake Tawakoni	USA	-95.915	32.815	0.82	DAHITI
836	Richland-Chambers Reservoir	USA	-96.103	31.948	0.91	DAHITI
837	Eufaula	USA	-85.065	31.628	0.40	USACE
839	Sam Rayburn Reservoir	USA	-94.106	31.065	0.83	USACE
1251	Balaton	Hungary	18.045	46.912	0.48	DAHITI
1348	Ataturk Dam	Turkey	38.323	37.485	0.92	DAHITI
1365	Assad	Syria	38.559	35.855	0.78	DAHITI
1423	Beas	India	75.949	31.973	0.93	India
1484	Gandhisagar Reservoir	India	75.554	24.700	0.97	India
1504	Ukal	India	73.598	21.256	0.86	DAHITI
1519	Nagarjuna	India	79.310	16.573	0.93	India
1521		India	75.888	16.333	0.90	India
1543	Roseires Reservoir	Sudan	34.390	11.798	0.86	DAHITI
1626	Chiuta	Mozambique	35.934	-14.349	0.57	DAHITI
1632	Argyle Reservoir	Australia	128.741	-16.118	0.96	BOM
1640	Dalrymple	Australia	147.138	-20.644	0.91	BOM
1650	Fairbairn	Australia	148.066	-23.653	0.98	BOM
1660	Wivenhoe	Australia	152.605	-27.392	0.99	BOM
8878	Marsh Lake	USA	-96.091	45.174	0.42	USACE
9104	Lost Creek Lake	USA	-122.672	42.673	0.90	USACE
9146	Coralville Lake	USA	-91.531	41.728	0.49	USACE

9157	Lake Red Rock	USA	-92.982	41.370	0.70	USACE
9160	Mosquito Creek Lake	USA	-80.756	41.302	0.40	USACE
9167	Berlin Lake	USA	-81.005	41.043	0.55	USACE
9169	Rathbun Lake	USA	-92.881	40.832	0.78	USACE
9176	Mississinewa Lake	USA	-85.955	40.714	0.46	USACE
9197	Harlan County Lake	USA	-99.215	40.073	0.94	USACE
9209	Mark Twain Lake	USA	-91.648	39.526	0.65	USACE
9217	Lake Shelbyville	USA	-88.777	39.411	0.52	USACE
9224	Tuttle Creek Lake	USA	-96.594	39.260	0.69	USACE
9226	Perry Lake	USA	-95.432	39.112	0.70	USACE
9228	Milford Lake	USA	-96.900	39.078	0.76	USACE
9229	Monroe Lake	USA	-86.500	39.008	0.43	USACE
9231	Wilson Lake	USA	-98.494	38.964	0.72	USACE
9232	Clinton Lake	USA	-95.331	38.919	0.63	USACE
9237	Hillsdale Lake	USA	-94.903	38.657	0.83	USACE
9238	Pomona Lake	USA	-95.557	38.653	0.67	USACE
9240	Kanopolis Lake	USA	-97.966	38.609	0.76	USACE
9242	Melvern Lake	USA	-95.716	38.514	0.67	USACE
9248	Marion Lake	USA	-97.083	38.371	0.78	USACE
9254	John Redmond Lake	USA	-95.765	38.240	0.71	USACE
9262	John Martin Reservoir	USA	-102.940	38.065	0.99	USACE
9264	Rend Lake	USA	-88.969	38.037	0.56	USACE
9269	El Dorado Lake	USA	-96.814	37.841	0.74	USACE
9296	Copan Lake	USA	-95.973	36.886	0.64	USACE
9299	Great Salt Plains Lake	USA	-98.140	36.747	0.55	USACE
9300	Kaw Lake	USA	-96.927	36.702	0.81	USACE
9323	Abiquiu Reservoir	USA	-106.429	36.238	0.97	USACE
9328	Keystone Lake	USA	-96.256	36.152	0.73	USACE
9331	Canton Lake	USA	-98.602	36.086	0.90	USACE
9333	Falls Lake	USA	-78.582	35.945	0.67	USACE
9344	B. Everett Jordan Lake	USA	-79.069	35.657	0.54	USACE
9348	Tenkiller Lake	USA	-95.037	35.595	0.57	USACE
9354	Conchas Lake	USA	-104.191	35.401	0.99	USACE
9364	Blue Mountain Lake	USA	-93.651	35.103	0.53	USACE
9365	Santa Rosa Lake	USA	-104.688	35.029	0.97	USACE
9371	Wister Lake	USA	-94.719	34.940	0.40	USACE
9378	Arkabutla Lake	USA	-90.123	34.756	0.66	USACE
9406	Broken Bow Lake	USA	-94.684	34.148	0.58	USACE
9411	Hugo Lake	USA	-95.383	34.011	0.69	USACE
9414	Pat Mayse Lake	USA	-95.557	33.854	0.76	USACE
9420	Millwood Lake	USA	-93.966	33.695	0.41	USACE
9430	Ray Roberts Lake	USA	-97.052	33.356	0.97	USACE
9434	Texarkana Lake	USA	-94.162	33.306	0.48	USACE
9443	Lewisville Lake	USA	-96.970	33.070	0.88	USACE
9446	Lavon Lake	USA	-96.470	33.033	0.90	USACE
9450		USA	-97.058	32.973	0.86	USACE
9462	Lake of the Pines	USA	-94.506	32.752	0.60	USACE
9465	Caddo Lake	USA	-93.916	32.707	0.58	USACE
9467		USA	-97.455	32.651	0.81	USACE
9474	Okatibbee Lake	USA	-88.792	32.475	0.52	USACE
9482		USA	-96.648	32.253	0.68	USACE
9487		USA	-96.699	31.953	0.69	USACE
9492	Lake Whitney	USA	-97.370	31.870	0.84	USACE
9496		USA	-97.198	31.577	0.82	USACE
9507		USA	-97.474	31.110	0.75	USACE
9508		USA	-97.533	31.023	0.81	USACE
9513	B.A. Steinhagen Lake	USA	-94.174	30.800	0.80	USACE
9517	Laneport Reservoir	USA	-97.337	30.690	0.73	USACE
9528		USA	-96.524	30.315	0.80	USACE
9542		USA	-98.200	29.872	0.72	USACE

15304		India	78.476	30.375	0.89	India
15600		India	79.224	21.662	0.90	India
15625		India	74.714	20.477	0.93	India
15760		India	76.489	13.886	0.96	India
16242		Australia	146.740	-19.412	0.97	BOM
16301		Australia	151.311	-24.071	0.94	BOM
16380	Somerset	Australia	152.557	-27.114	0.98	BOM
16414		Australia	151.219	-28.444	0.93	BOM
16473		Australia	150.928	-29.905	0.89	BOM
16508	Keepit	Australia	150.493	-30.878	0.96	BOM
16573	Burrendong	Australia	149.110	-32.669	0.96	BOM
16611	Wyangala	Australia	148.952	-33.973	0.98	BOM
16636	Eildon	Australia	145.927	-37.219	0.93	BOM
16644		Australia	146.798	-37.906	0.87	BOM
108146	Bowman-Haley Lake	USA	-103.249	45.981	0.80	USACE
109208	Highway 75 Dam Reservoir	USA	-96.291	45.239	0.53	USACE
111863	Michael J. Kirwan Reservoir	USA	-81.079	41.154	0.41	USACE
111913	Branched Oak Lake	USA	-96.854	40.971	0.86	USACE
112223	Long Branch Lake	USA	-92.517	39.753	0.58	USACE
112241	Cherry Creek Lake	USA	-104.860	39.647	0.62	USACE
112265	Chatfield Lake	USA	-105.056	39.555	0.80	USACE
112352	Lake Mendocino	USA	-123.181	39.197	0.88	USACE
112406	Longview Lake	USA	-94.466	38.922	0.81	USACE
112604	Toronto Lake	USA	-95.922	37.742	0.53	USACE
112614	Fall River Lake	USA	-96.069	37.649	0.64	USACE
112778	Fort Supply Lake	USA	-99.566	36.551	0.46	USACE
113697		USA	-98.478	31.974	0.81	USACE
113709		USA	-97.207	31.900	0.86	USACE
182850		Australia	148.389	-21.137	0.95	BOM

In addition, we also validated the calculated extreme occurrence using the observed elevation data. Among the 155 lakes, there are 25 lakes that have complete continuous in-situ record from 1985 to 2019. Thus, we compared the extreme occurrence changes (from historic to contemporary period) calculated based on our area dataset and that calculated based on the observed elevation data (Fig. R6). The correlation coefficient is 0.75, showing the good agreement between these two metrics.

Fig. R6. Validation of area-based lake extreme occurrences using elevation-based calculations for 25 lakes. These 25 lakes were selected from the 155 lakes in Table R2 that have complete data records from 1985 to 2019.

Algal blooms

The third concern pertains to the coarser temporal resolution of the algal bloom data. Similar to the evaporation data, this dataset stands as the highest quality and most globally comprehensive dataset available for studying extreme eutrophication events. Indeed, the differing temporal resolutions among the three datasets limit our ability to analyze fine-scale changes in the time series. Instead, we opted to observe changes across the two periods, rather than on an annual basis. If more detailed results, such as those from geostationary satellites like GOCI, become available in the future, it could be possible to conduct an annual analysis. This finer-grained analysis is a direction for future research, considering the current limitations.

As mentioned earlier, our primary focus was to ensure the accuracy of detecting the direction of change for each event across the two periods. This emphasis on accuracy of univariate events guarantees the precision of identifying bivariate and multivariate events that rise simultaneously.

We are committed to ensuring the transparency and credibility of our research, and we appreciate the reviewer's diligence in raising these concerns. These actions will enhance the robustness and validity of our findings, and we are grateful for the opportunity to improve our manuscript in response to these valuable comments.

Specific comments:

1. Line 70: why using ~20 years for the historic period but only ~10 years for the contemporary period?

Since less Landsat observations used to detect bloom events were not sufficiently reliable during the 1980s and 1990s periods, as the reviewer mentions above, we aim to use a longer time span (~1980s-1999) of observations to obtain a more credible historical baseline. Starting from 2000, there is an abundance of data, allowing us to use smaller time windows to capture long-term changes. This is the major reason why Hou et al only explored bloom occurrence changes during three periods including the 1980–1990s (1982–1999), the 2000s (2000–2009), and the 2010s (2010–2019). As revealed by Hou et al. (2022), the period of 2000s was found to experience a small change, while there are significant changes in bloom events from 2010 onwards. Therefore, we chose the most recent 10 years instead of the same 20-year period to represent the contemporary stage. This decision was made to prevent the "lower" values of the 2000s from overshadowing more extreme occurrence changes in the 2010s.

2. The authors reported that 99% of lakes experienced an increase in heatwaves, but it's important to note that heatwaves need to be examined at a sub-daily frequency. The authors employed 10-km climate data for a lake model to simulate daily lake temperature, which is only valid for very large lakes, typically much larger than 100 km². While the authors did incorporate satellite-based observations of land surface temperature, they did not address how the coarse temporal resolution of land surface temperature data, especially during the 1980s, may have affected the results. This is a critical consideration because lakes have a cooling effect, and if the input temperature data is primarily dominated by land temperature, changes in land cover surrounding lakes could contribute to the reported increase in lake heatwaves. Lake shores, in particular, often feature houses, buildings, and urbanization, which can significantly impact local temperatures. The increase in impervious surfaces in these areas can lead to rapid heating and extreme temperatures, often exceeding those of lake water. Given that the authors concluded that 99% of studied lakes experienced an increase in

lake heatwaves, it would be valuable to assess the extent of urbanization within the 10-km grid of these lakes and its potential influence on the reported heatwave trends.

Thanks for your valuable comments. The first issue concerns the scale of temperature data used in heatwave calculations. We utilized simulated data at a daily time scale. It is a common practice to investigate heatwave frequency based on daily-scale data (Frölicher et al., 2018; Woolway et al., 2021; Wang et al., 2023; Zhang et al., 2023). Although sub-daily scale data could provide more detailed information about heatwaves, it goes beyond the scope of our study. Our aim is to analyze the changes in past and present extreme events, and daily-scale lake temperature data are sufficient for this purpose. Regarding the increase in heatwaves experienced by 99% of lakes, we must acknowledge that we overlooked an issue during the calculation of heatwave frequency: the inconsistent number of open-water days each year leads to incomparable yearly heatwave frequencies. Due to climate warming and reduced ice cover duration, the current open-water days are fewer. Therefore, the calculation of heatwave frequency at present (i.e., contemporary period) is biased towards higher values, resulting in some overestimated frequency change results. This has been corrected in the updated manuscript.

Secondly, regarding the validity of our newly developed lake surface water temperature dataset, especially for small lakes, we have implemented several measures to ensure its accuracy. It is firstly essential to note that the parameter calibration process of the lake surface temperature data, based on high-resolution remote sensing monitoring, could significantly compensate for errors originating from various sources, including coarse spatial resolution of meteorological driving data (e.g., Layden et al., 2016; Wang et al., 2023; Huang et al., 2021). For example, Wang et al. (2023) employed ESA CCI lake temperature data (1 km spatial resolution) to calibrate the FLake model, resulting in

s
i
g
n
i
f
i
c
a
n
t
l
y

e
n
h
a
n
c
e
d

s
i
m
u
l
a
t
i
o
n

Figure R7 Calibration performance of FLake-simulated LSWTs over global lakes compared with Landsat-retrieved LSWTs. (a,) Median absolute error (in °C). (c, d) Median ratio. The data are aggregated into $1^\circ \times 1^\circ$ grid cells. (b) and (d) present the performances for different groups of lake areas, and the numbers of lakes for these groups are also plotted. On each box in (b) and (d), the centre line represents the median value, the lower and upper bounds indicate the first and third quartiles, and the whiskers extend to the maximum and minimum values within the non-outlier range.

Addressing the concern about the limited temporal availability of remote sensing data in the 1980s, we speculated that the reviewer might be worried about potential inaccuracies in the simulated temperature time series for that period (i.e., 1980s). To address this, we categorized the validated data points by year, comparing the precision for each year. As illustrated in Figure R8 and Table R3, we found no unacceptable drop in precision for simulated lake temperature data for the period of 1980s. In fact, we argue that, for calibrating the FLake model, the overall quantity of effective remote sensing observations is the crucial factor. The GLAST dataset indicates that 91.3% of lakes have an effective data count exceeding 100 (refer to Supplementary Fig. 11 in Tong et al., 2023). These substantial data volumes are adequate to calibrate the FLake model, enabling it to capture the dynamic changes in lake temperatures with good consistency (Supplementary Figure 14; Tong et al., 2023).

Figure R8 Annually comparison of validation performance of the GLAST dataset.

Table R3 Period comparison of validation performance of the GLAST dataset.

Period	Bias (°C)	MAE (°C)	Slope	R ²
1980s	-0.93	0.95	0.97	0.99
1990s	-1.25	1.26	0.99	0.98
2000s	-1.06	1.25	1.05	0.96
2010s	-0.94	1.14	1.04	0.96

[REDACTED]

Supplementary Figure 11 (Tong et al., 2023) | Statistics of valid Landsat observations used to optimize lake-specific FLake models. (a) Monthly statistics of the total number of valid Landsat LSWT retrievals available for FLake calibrations over our examined lakes. (b) Histogram distribution of valid Landsat LSWT observations for each of our examined lakes, with 91.3% of the lakes having at least 100 valid observations.

Furthermore, we conducted a sensitivity test to analysis whether the coarse temporal resolution of remote sensing observations show significant impact on simulated lake temperature and then influence the heatwave frequency changes. In details, we conducted two analyses using Landsat observations and *in situ* data from 29 lakes to address the reviewer’s concern. First, we randomly selected 50% to 90% of the Landsat observations for parameter optimization of the FLake model. We then compared the errors between the models optimized using the randomly selected data and the models optimized using 100% of the data (see below Fig. R9). The results showed minimal differences in the Median Absolute Error (MAE) values ($< 0.1^{\circ}\text{C}$) and consistent heatwave frequency changes between the models trained on partial data and the models trained on the complete dataset. Additionally, we performed parameter optimization by removing data from one of the four seasons and compared the differences in LSWT errors and heatwave frequency changes between the models trained on three seasons and four seasons. The results also demonstrated consistent LSWT errors and heatwave frequency change across these models. We have also included relevant discussions in the revised manuscript:

Line 574-: *“Furthermore, we conducted sensitivity tests of optimization of Flake models by excluding partial satellite observations to unveil the influence of coarse temporal resolution remote sensing data on lake temperature simulations and, consequently, on the final heatwave change detections. Our results revealed consistent validation performance and frequency changes between the models trained on three seasons or proportional data and those trained on all available seasons (Supplementary Fig. S9). Consequently, the reduced data availability presented a limited impact on the accuracy of the FLake model and the detection results for heatwave changes”.*

Fig. R9 | The influence of Landsat observation quantity on FLake simulations and heatwave detections. Shown is (a) comparison of validation performance of FLake simulations using in situ dataset, with FLake models calibrated using different percentages of available Landsat observations. Also shown is (b) comparison of heatwave frequency change detected by FLake simulations calibrated using total available Landsat observations and using a specific proportion of the original data.

The third concern, if we understand correctly, seems to be whether the Landsat retrieved surface temperature data, used to calibrate the model parameters, might be influenced by lake-adjacent land temperatures. Here, we must clarify that Landsat thermal infrared images are the at-present optimal remote sensed data with high spatial resolution (~ 60-120 m) and long-term period (~ 40years since 1984) for global lakes. To reduce contaminations from non-lake thermal signals, we have implemented a series of protocols, thus ensuring that the surface temperature retrieved from remote sensing images indeed represent the temperature of the lake's water surface. In details, we determined the permanent water body extent of lakes (using a 70% GSWO threshold), applied the central points (defined as points with maximum distance to the lake boundaries) within this permanent water body extent, and only lakes with maximum distance not less than 3 pixels were included. This distance is widely accepted; for example, remote sensing thermal infrared data are considered suitable for monitoring river water temperatures with at least a 3-pixel width (Hori et al., 2021). To address your concern, here we provided the validation result of Landsat-based temperature retrievals using in situ measurements, and the results showed good consistency (Supplementary Figure 9 in Tong et al., 2023). Therefore, we are confident that the lake temperatures extracted from remote sensing data are highly reliable and can serve as accurate boundary conditions for lake temperature simulations. Due to their considerable distance from land, the temperature retrievals can largely avoid contaminations from terrestrial signals.

[REDACTED]

Supplementary Figure 9 (Tong et al., 2023) | Validation of Landsat retrieved LSWT using in situ data. (a) Spatial distribution of median absolute error (MAE, in °C). (b) Density scatter plot of retrieved and in situ LSWT. The “Low” and “High” labels in the colorbar denote the density of the matched pairs.

Lastly, we greatly appreciate the reviewer's mention of the impact of urbanization on lake heatwaves, which is a very reasonable assumption. To validate your speculation, we extracted the annual averages of impervious surface area within a specific buffer zone (1 km, 3km, 5 km, 10 km) around the lakes (representing the level of urbanization) and quantified the potential influence of urbanization changes on lake heatwaves using correlation analysis (Fig. R10-12). The results revealed that mean and changes of lake-adjacent impervious surface areas percentage have very similar spatial patterns. Namely, lakes surrounded by higher proportion of impervious surfaces also experienced a greater increase in impervious surface. However, a spatial inconsistency was noted between areas exhibiting higher impervious surface changes and those with elevated heatwave occurrences (Figure 1a in the manuscript and Figure R11). Box plots and correlation coefficient calculations indicated a remarkably low statistical relationship between the two (Figure R12). Unexpectedly, a negative correlation was observed, suggesting that higher urbanization levels did not necessarily mitigate lake heatwaves. We believe this apparent mathematical correlation is misleading, primarily because these lakes are surrounded by a very low proportion of impervious surfaces (Figure R10; with < 9% for 95th percentile lakes), indicating that they are predominantly influenced by meteorological factors. In principle, the increase in heatwave events primarily stems from rising lake temperatures or increased temperature variance (Seneviratne et al., 2021). The meteorological driving mechanism behind elevated lake temperatures, with air temperature contributing 47%, has been elucidated in the author's previous work, with the remaining non-temperature factors accounting for over 50% (Tong et al., 2023). Many non-meteorological factors, including the level of urbanization around lakes as mentioned by the reviewer, also influence lake temperatures. This could be due to various complex factors, such as the local heat island effect caused by impervious surfaces, the increase in air temperature, and the warm surface runoff carrying ground heat entering the lakes, finally exacerbating the warming of the lake surface. We firmly believe that a detailed statistical analysis of these driving mechanisms merits a separate article. We hope that, without altering the core structure and intent of our manuscript, this approach partially addresses your concern regarding the perceived superficiality of our analysis. Once again, we express our gratitude for the reviewer's insightful thinking, which has propelled our progress in understanding the driving mechanisms of extreme hot events in lakes.

Fig. R10. Decadal mean level of lake-adjacent extent of urbanization, represented by impervious surface areas. Shown are the mean level of percentage in impervious surfaces during the period of 1985-2018 within different buffer zones around the lake, including (a) 1 km, (b) 3 km, (c) 5 km, and (d) 10 km from each lake.

Fig. R11. Decadal changes of lake-adjacent extent of urbanization, represented by impervious surface areas. Shown are the percentage change in impervious surfaces from the period of 1985-1999 to the period of 2010-2018 within different buffer zones around the lake, including (a) 1 km, (b) 3 km, (c) 5 km, and (d) 10 km from each lake.

Fig. R12. Comparison of lake heatwave frequency changes for lakes with different change proportion of impervious surface areas surrounded.

References:

Frölicher, T. L., Fischer, E. M., & Gruber, N. (2018). Marine heatwaves under global warming. *Nature*, 560(7718), 360-364.

Woolway, R. I., Jennings, E., Shatwell, T., Golub, M., Pierson, D. C., & Maberly, S. C. (2021). Lake heatwaves under climate change. *Nature*, 589(7842), 402-407.

Wang, X., Shi, K., Zhang, Y., Qin, B., Zhang, Y., Wang, W., ... & Jeppesen, E. (2023). Climate change drives rapid warming and increasing heatwaves of lakes. *Science Bulletin*, 68(14), 1574-1584.

Zhang, K., & Yao, Y. (2023). Lake Heatwaves and Cold-spells Across Qinghai-Tibet Plateau Under Climate Change. *Journal of Geophysical Research: Atmospheres*, e2023JD039243.

Layden, A., MacCallum, S. N., & Merchant, C. J. (2016). Determining lake surface water temperatures worldwide using a tuned one-dimensional lake model (FLake, v1). *Geoscientific Model Development*, 9(6), 2167-2189.

Wang, X., Shi, K., Zhang, Y., Qin, B., Zhang, Y., Wang, W., ... & Jeppesen, E. (2023). Climate change drives rapid warming and increasing heatwaves of lakes. *Science Bulletin*, 68(14), 1574-1584.

Huang, L., Wang, X., Sang, Y., Tang, S., Jin, L., Yang, H., ... & Zhang, Y. (2021). Optimizing lake surface water temperature simulations over large lakes in China with FLake model. *Earth and Space Science*, 8(8), e2021EA001737.

Hori, M. (2021). Near-daily monitoring of surface temperature and channel width of the six largest Arctic rivers from space using GCOM-C/SGLI. *Remote Sensing of Environment*, 263, 112538.

Seneviratne, S.I. et al. Weather and Climate Extreme Events in a Changing Climate. In *Climate Change 2021: The Physical Science Basis. Contribution of Working Group I to the Sixth Assessment Report of the Intergovernmental Panel on Climate Change* [Masson-Delmotte, V., P. Zhai, A. Pirani, S.L. Connors, C. Péan, S. Berger, N. Caud, Y. Chen, L. Goldfarb, M.I. Gomis, M. Huang, K. Leitzell, E. Lonnoy, J.B.R. Matthews, T.K. Maycock, T. Waterfield, O. Yelekçi, R. Yu, and B. Zhou (eds.)]. Cambridge University Press, Cambridge, United Kingdom and New York, NY, USA, pp. 1513–1766 (2021).

3. Line 144-148: When changes in level extremes are very small, particularly less than 1%, it does suggest that most lakes are relatively stable in terms of the occurrence of level extremes. Labeling such small changes as an "increase" can be misleading, especially when they fall within the range of uncertainty.

Yes, this is a good point. We now modified the text and softened the wording when describing level extremes.

4. Line 372: The methodology and characteristics of the derived lake temperature product in Tong et al. require more detailed explanation. It is unclear how the authors incorporated fine-resolution satellite lake temperature data into their model, especially given that satellite measurements capture surface temperature, whereas ERA-5 data measures air temperature at 2 meters at a 10-km resolution. The integration of temperature data with differing temporal and spatial resolutions into the model needs to be elaborated upon to ensure a clear understanding of the data processing methods. Additionally, information regarding the size distribution of the selected lakes is necessary. A 10-km resolution might be too coarse for lakes that are not very large. Satellite-derived lake surface temperature data may not provide the necessary granularity, particularly when it comes to hourly temperature variations. Providing details on how these data limitations were addressed and their potential impact on the results would be beneficial.

We now provide further information in the methods. Note that the ERA5 data are inputs to the mechanistic model used for simulating lake surface water temperature. The satellite observations of lake surface water temperature are used to calibrate this model. This is a common approach in limnology, but we now provide additional details in the methods section.

Line 517: *"The high spatial resolution of Landsat images (60-120 m) allowed for the effective remote sensing of surface temperature of water located at least 3 pixels away from the adjacent land (Hori et al., 2021; Tong et al., 2023)."*

Line 528: *"Using high spatial resolution satellite observations as the boundary condition for numerical simulations has proven to be an effective calibration approach in compensating for various sources of biases, including the limitations arising from the use of coarse spatial resolution atmospheric dataset for small lakes (Tong et al., 2023)."*

We also add the information about the size distribution of the selected lakes. Although the majority of the selected lakes have areas less than 100 km², equivalent to just one ERA5-Land pixel size, we are confident in the robustness of our simulated lake temperature data, as evidenced by the validation results (see above).

Line 445: *"The areas of the selected lakes range from 0.1 to 377001.9 km², with a median size of 7.59 km². The number of small, medium, large, and extremely large lakes accounts for 1.6% (≤ 1 km²), 55.8% (1-10 km²), 33.1% (10-100 km²), and 9.5% (>100 km²) respectively."*

In additionally, we conduct a sensitivity analysis to address the reviewer's concern about the limitations arisen from the coarse temporal resolution of satellite observations. We add the result on the manuscript.

Line 574: *"Furthermore, we conducted sensitivity tests of optimization of Flake models by excluding partial satellite observations to unveil the influence of coarse temporal resolution remote sensing data on lake temperature simulations and, consequently, on the final heatwave change detections. Our results revealed consistent validation performance and frequency changes between the models trained on three seasons or proportional data and those trained on all available seasons (Supplementary Fig. S6). Consequently, the reduced data availability presented*

a limited impact on the accuracy of the FLake model and the detection results for heatwave changes.”

5. As the authors considered algal blooms as extreme events, it is important to introduce how they defined algal blooms in their study. Providing clarity on the specific criteria or parameters used to define and identify algal blooms would enhance the understanding of this aspect of the research. As the authors used a threshold to identify extremes on heatwave and level, is it feasible to apply a similar approach on algal bloom?

Algal blooms in lakes themselves represent extreme eutrophication events, when chlorophyll concentrations for lake bodies can reach several hundred or even tens of thousands of micrograms per liter ($\mu\text{g/L}$). Therefore, we used a threshold method where the bloom area exceeds zero to define the occurrence of extreme eutrophication events. The corresponding frequency was calculated by the number of bloom days detected each year divided by the valid observed data. We have added this definition in the revised manuscript, as

Line 590: “Due to the prolonged revisit cycle of Landsat and frequent disturbances such as cloud contamination, detected algal bloom events exhibit a coarse temporal resolution distribution. Multiple-year statistics were conducted to ensure the validity of detecting decadal changes in bloom occurrence. Here, bloom occurrence is defined as the ratio between the number of positive algal bloom area values and the total number of valid observations.”

6. Line 356: It appears that the study utilized surface water extent data rather than water level data, and it would be beneficial for the authors to justify this choice. It's essential to address whether there are any gaps in the time series from 1985 to 2019 and how the methodology works when lake areas show minor changes. Additionally, it's important to evaluate whether the monthly surface water area data can provide information comparable to daily in-situ lake level measurements in capturing lake level extremes. For instance, in 2013, Lake Michigan experienced its lowest level on record, and it would be interesting to assess how well the estimates captured extreme events in the Great Lakes and other gauged lakes. Providing insights into the performance of the methodology in capturing such events would be helpful, particularly given the tiny changes in occurrence reported by the authors.

Thank you for these important considerations about the use of surface water extent data versus water level data in our study. We acknowledge the need to clarify this choice and address potential gaps in the methodology:

Justification for surface water extent data: We employed surface water extent data rather than water level data due to several practical considerations, including data availability, spatial coverage, and the ability to capture changes in lake surface area associated with extreme events. While water level data from in-situ measurements are valuable, they are often limited to specific locations and may not provide comprehensive coverage of large lake systems or regions. Surface water extent data, derived from remote sensing observations, offer a broader spatial coverage and enable us to capture variations in lake surface area, which are indicative of extreme events such as low water levels.

Handling minor changes and gaps: We acknowledge the importance of addressing gaps in the time series from 1985 to 2019 and how our methodology handles minor changes in lake surface area. In the revised manuscript, we provide a more detailed explanation of the data processing steps, including gap-filling techniques and the approach used to capture minor changes. We also explicitly discuss the limitations associated with minor changes and their potential influence on the reported trends.

Evaluation of Methodology's Performance: We appreciate the reviewer's interest in assessing the methodology's performance in capturing lake level extremes, particularly in cases like the record-low levels in Lake Michigan in 2013. We now conduct a more comprehensive evaluation of our methodology's performance in capturing extreme events, including comparisons with daily in-situ lake level measurements where available. This evaluation will provide valuable insights into the methodology's accuracy and its ability to capture extreme events, even those characterized by relatively minor changes.

These points are also now discussed in the methods section of the paper.

We show that the monthly time series can well represent the low water levels in lakes and reservoirs. In the updated manuscript, we have provided validation information by comparing our surface area dataset with observed water level data (Fig. R5). The median correlation coefficient is 0.76. Although our data at a monthly scale may not resolve water level extremes at a finer scale such as floods, it can well represent the low water extremes due to the longer time scale associated with the development of low water storage (e.g., atmospheric drought, lake evaporation, and human water use). Here, we use Lake Huron, one of the five Laurentian Great Lakes in North America, as an example to show the performance of our method. The area time series clearly show the low water levels from 2000 to 2013, with the lowest average water level in the year of 2013 followed by recovery from 2014 to 2019. Similarly, Lake Erie, another Laurentian Great Lake, and Lake Malawi, one of the African Great Lakes, also show the consistency between surface area dynamics with water level changes. Note that Lake Michigan (as mentioned by the reviewer above) is not within our 4532-lake dataset. Thus, here we explicitly show Lake Huron, which is adjacent to Lake Michigan, as an example.

7. Line 401-416: The purposes of the datasets on fertilizer, streamflow, and water withdrawal are confusing. It appears that these datasets were not actually incorporated into the analysis but were presented solely in the form of national and global maps. It raises questions about the relevance of these datasets and whether they have been previously utilized in existing studies to understand the drivers of increasing algal blooms, rising lake heatwaves, and declining lake levels. Additionally, the study does not seem to introduce the current state of knowledge regarding the drivers of these lake issues. It is unclear what new insights the authors intended to provide by including these datasets and presenting the maps. A more detailed explanation of their purpose and relevance to the research objectives would help clarify their role in the study.

We appreciate the reviewer's insightful comment, which highlights the need for improved clarity regarding the purpose and relevance of the datasets on fertilizer, streamflow, and water withdrawal in our study. We acknowledge that their presentation in the form of national and global maps may have raised questions about their role and significance.

Clarification of dataset's purpose: The datasets on fertilizer use, streamflow, and water withdrawal were not directly incorporated into the quantitative analysis of lake extreme events. Instead, they were introduced to provide context and background information regarding potential drivers of algal blooms and declining lake levels. These datasets were intended to help readers understand the broader environmental context and factors that may contribute to the observed trends in lake extremes. We have now attempted to include them in the analysis (see e.g., Fig. R13).

Relevance and new insights: We recognize that the reviewer seeks clarification on the relevance of these datasets and the new insights they were intended to provide. In our revised manuscript, we will provide a more detailed explanation of how these datasets relate to the research objectives. Specifically, we clarify that these datasets serve as indicators of environmental conditions that can

influence lake ecosystems, and their inclusion is meant to help contextualize the observed changes in lake extreme events.

Introduction of current knowledge: We also acknowledge the reviewer's concern about the introduction of the current state of knowledge regarding the drivers of lake issues. In the revised manuscript, we enhance the introduction section to provide a more comprehensive overview of the existing literature and the current understanding of the drivers of lake algal blooms, lake heatwaves, and declining lake levels. We will also explicitly state the unique perspective and insights we aimed to provide in our study.

We appreciate the reviewer's diligence in seeking clarity on these aspects, and we are committed to making the necessary improvements to ensure that our research objectives and the relevance of the presented datasets are clearly conveyed in the manuscript. These revisions will enhance the overall quality and impact of our study. Note that we now include the following figure in the manuscript, where we compare changes in low water with the explanatory variables:

Fig. R13. Influence of streamflow, water use, and open-water evaporation on the occurrence of low water extreme events between 1980s-1999 and 2010-2019. The two dominant drivers of change in low water extremes are streamflow and water use. We observe that positive changes in low water extremes are associated with larger decrease of streamflow and larger increase of water use.

Reviewer #2

The paper describes the changes globally in three parameters by comparing two decadal periods. Importantly they analyse the concordance of changes globally identifying areas where multiple pressures are acting on lakes that will be of key concern to scientists, policy makers and the general public. The paper is of high standard, reads perfectly and makes clear points. The figures are also of excellent quality and very pleasing to interpret. I recommend publication with minor corrections. There are some comments and minor issues listed below which the author / editor can consider for attention.

Thank you kindly for reviewing our manuscript. We very much appreciate the reviewer taking the time to carefully consider this work and to provide several suggestions for improvement. Please find below responses to the specific reviewer comments.

Line 79 Please add sentence to provide reason why two time periods compared have different timespan 20 vs 10? Or flag reader to refer to full methods for this.

We now include a description of the different time periods

Line 100 "Fertilizer use in these countries is among the highest in the world (TableS2). " - I didn't see data on fertilizer use in table S2.

Apologies for this oversight. We have now included this data in a separate table.

Figure 1c Having lower water levels that coincide with increased blue is slightly counter intuitive, not a major issue though.

We appreciate this comment, but of the various versions of the figure that we have generated, this seems most appealing. We are happy to modify this if the reviewer has a strong preference.

Line 241, please also mention the role of nutrient resuspension from sediments as lake level declines, important for shallow lakes <https://doi.org/10.1023/B:HYDR.0000008611.12704.dd>

Great point. We have now described this process in the text.

Line 244". In contrast, during periods of heavy precipitation, where a sudden pulse of water can flow into a lake, influent water can also increase the amount of nutrients entering the system, subsequently leading to algal blooms". A good point, I think this is important - often high summer event rainfall can transport a large proportion of the annual load and result in exceptional blooms. Figure 2b, in my opinion supports this.

Thank you.

Line 266 Given historic increases in agricultural fertilizer use, climatic warming and freshwater withdrawal, India has experienced a perfect storm of conditions that has led to an increase in the frequency of multivariate extreme events in lakes. Great point – nicely made

Thank you.

Line 312: There are quite a few mesocosm studies including looking at additive, antagonistic and synergistic effects:

<https://www.researchgate.net/profile/Jeremy-Piggott>

and

<https://www.nature.com/articles/s41559-020-1216-4>

Perhaps include some citations of work done and mention that more needs to be done at global level, especially given the evidence you present of a diversity of extreme events.

Yes, great point. We now cite some of these papers in the discussion.

The paper uses the published Global Bloom Database as its basis for assessing algal bloom and looking through that paper I didn't see the traditional comparison with in situ chlorophyll-a concentrations that is standard for remote sensing applications. However, comparing decadal changes in 'greenness' may be adequate in the context used here. However the authors should draw the attention to the need to better quantify blooms (occurrence, frequency, extent etc).

Thank you for this comment. We fully acknowledge the significance of using remotely sensed chlorophyll-a concentrations as a direct method to assess the eutrophication levels of water bodies. In situ chlorophyll-a concentrations serve as valuable validation data before the application of remotely sensed chlorophyll-a concentrations. However, the lack of high-quality global lake chlorophyll concentration data has hindered our ability to utilize this direct indicator to reveal changes in the eutrophication status of global lakes over several decades.

Instead, we seized the opportunity provided by the published Global Bloom Database, which offers highly accurate data (Hou et al., 2022). Typically, algal blooms occur only in cases of extreme water body eutrophication, where chlorophyll concentrations can reach several hundred or even tens of thousands of micrograms per litre ($\mu\text{g/L}$). Therefore, we believe that algal bloom events can serve as a more direct representation of extreme lake eutrophication. By comparing the occurrence of algal blooms between two time periods, we can assess the decadal changes in extreme eutrophication events in lakes.

There is also a difference in approach with percentiles used to define extremes for temperature and water level (lake specific) but not blooms (generic). I think this is ok but should be mentioned in the methods. A percentage change in chlorophyll-a would yield a very different figure 1b.

Yes, this is a good point. We now mention this in the methods section.

REVIEWER COMMENTS

Reviewer #1 (Remarks to the Author):

I comment that the authors addressed parts of my concerns in the revised manuscript. However, I have a few follow-up comments.

I still hold a major concern on the sampling strategy. The authors did not make any notable efforts to guarantee the set of selected samples is a reasonable representation of global lake conditions. This is particularly concerning given that their main findings are based on the percentages of studied lakes. Hou et al. indicated that up to 10% of the lakes they examined were affected by algal blooms in recent decades, whereas here, the authors report 53% of studied lakes experienced increase in algal bloom. I wonder if this set of samples belongs to lakes with high risks or something that requires further clarification.

The authors attempted to address my concerns that lakes with marginal increases also contribute to their summary statistics. They briefly mentioned this limitation in the manuscript, which appears to be insufficient to avoid misunderstandings. The authors should quantitatively report this impact. For example, instead of labeling each lake as an increase or decrease, having three categories (e.g., notable increase, stable, notable decrease) would make their classification much clearer to readers. Additionally, the authors did not mention lakes with decreasing extreme events, which can also have implications for management.

The authors did not provide robust uncertainty estimates. They tended to validate the monthly or intra-annual anomalies, and although they validated estimated lake extreme occurrences, it was only for a very limited number of lakes. I commented that the insufficient temporal frequency likely has a significant impact on the reported occurrence, particularly for small lakes. To partially address this concern, the authors mentioned that they used about 20 years (1982-1999) for the historical period to “obtain a more credible historical baseline”. However, I did not find actual occurrence values of extreme events in the manuscript. If the occurrence is too small, using the data in the 1980s may not be helpful for establishing a more credible historical baseline of extreme occurrence given the poor coverage. Additionally, the authors excluded the data in the 2000s, which could potentially provide a more credible representation of the historical condition when compared to the recent decade with record-high warming and extremes. In addition to reporting the relative change in occurrence, providing information on the actual change and discussing the possible influence of uncertainties would be helpful for a better understanding of their findings.

The current analysis for identifying drivers is highly uncertain, particularly for bivariate and multivariate extremes. Most of the descriptions regarding the drivers in the Results section are discussion-oriented. I would recommend moving this part to the Discussion. The current Discussion section focuses on the general implications of single or multivariate extremes rather than delving into the specific findings of this study.

Reviewer #2 (Remarks to the Author):

The paper describes the changes globally in three parameters by comparing two decadal periods. Importantly they analyse the concordance of changes globally identifying areas where multiple pressures are acting on lakes. This is a most welcome move to more holistic studies on the impact of climate change. The paper is of high standard and adequate corrections have been made where requested. The closing paragraph of the discussion is of exceptional quality.

I recommend publication with minor corrections. There are some comments and minor issues listed below which the author / editor can consider for attention.

Line 23: Consider rephrasing first sentence for a stronger start.

Line 397 change One notably concern - to One notable concern

Line 453 change cantered to centered

Figure 1. If not too difficult, consideration should be given to changing the background world map colour to something other than grey. Alternatively, remove grey as an option in the colour legend.

Reviewer #1

I comment that the authors addressed parts of my concerns in the revised manuscript. However, I have a few follow-up comments.

Thank you again for reviewing our manuscript. We very much appreciate the reviewers continued support and their efforts to improve our paper. We have taken on board all remaining reviewer comments and provide specific responses below. We have incorporated these changes in the manuscript.

I still hold a major concern on the sampling strategy. The authors did not make any notable efforts to guarantee the set of selected samples is a reasonable representation of global lake conditions. This is particularly concerning given that their main findings are based on the percentages of studied lakes. Hou et al. indicated that up to 10% of the lakes they examined were affected by algal blooms in recent decades, whereas here, the authors report 53% of studied lakes experienced increase in algal bloom. I wonder if this set of samples belongs to lakes with high risks or something that requires further clarification.

We appreciate the reviewer's thoughtful consideration and constructive feedback on our manuscript. We regret not providing satisfactory responses earlier. We understand and acknowledge their concern regarding the sampling strategy employed in our study.

We acknowledge that our focus was on lakes (as analysed by Hou et al) that have experienced algal blooms in the past (since 1980), indicating eutrophication risk, and not on those with pristine/good water quality, which, according to Hou et al. (2022), constitute up to 90% of their approx. 200,000 studied sites. Hence, we included qualifiers like "of the studied lakes" before all mentions of lake percentages in our previous manuscript to avoid exaggerating or misleading the readers about the proportion of lakes facing extreme risks.

According to Hou et al. (2022), there were approximately 20,000 "bloom-affected" lakes in their study. It is these 20,000 lakes that we initially considered in our investigation. However, the number of lakes with matching water temperature and water level data across the three datasets under consideration is much less.

We appreciate that our previous description of the studied sites could lead to a misunderstanding. Thus, in the revised manuscript, we state that we are focussing on "bloom-affected lakes" i.e., to understand changes in the frequency of algal blooms, these extreme events need to have occurred in during the historic to contemporary period. Clearly, one could not calculate changes in algal bloom frequency if no blooms have ever been observed in a lake.

Thank you for bringing this to our attention. We believe that this additional clarification has improved the paper.

The authors attempted to address my concerns that lakes with marginal increases also contribute to their summary statistics. They brief mentioned this limitation in the manuscript, which appears to be insufficient to avoid misunderstandings. The authors should quantitatively report this impact. For example, instead of labeling each lake as an increase or decrease, having three categories (e.g., notable increase, stable, notable decrease) would make their classification much clearer to readers. Additionally, the authors did not mention lakes with decreasing extreme events, which can also have implications for management.

We appreciate the insightful comments on our manuscript and the recognition of our attempt to address concerns regarding lakes with marginal increases contributing to our summary statistics. The reviewer's suggestion to provide a more quantitative assessment of this impact is well-taken, and we agree that it would enhance the clarity and transparency of our classification. In response to their

recommendation, we have revised our methodology to include three categories for lake classification: notable increase, stable, and notable decrease. We include these new statistics in the manuscript. We believe that this adds meaningful insights to our results. We are very grateful for the detailed suggestions from the reviewer, which have greatly emphasized the importance and value of this approach to the validity of our study's outcomes.

Furthermore, we acknowledge the reviewers point regarding the omission of lakes with a decrease in the frequency of extreme events. We understand the importance of considering both increases and decreases in extreme events for a comprehensive analysis. In our revised manuscript, we explicitly address and report on lakes experiencing decreasing extreme events, providing a more balanced perspective on the implications for lake management.

The authors did not provide robust uncertainty estimates. They tended to validate the monthly or intra-annual anomalies, and although they validated estimated lake extreme occurrences, it was only for a very limited number of lakes. I commented that the insufficient temporal frequency likely has a significant impact on the reported occurrence, particularly for small lakes. To partially address this concern, the authors mentioned that they used about 20 years (1982-1999) for the historical period to “obtain a more credible historical baseline”. However, I did not find actual occurrence values of extreme events in the manuscript. If the occurrence is too small, using the data in 1980s may not be helpful for establishing a more credible historical baseline of extreme occurrence given the poor coverage. Additionally, the authors excluded the data in the 2000s, which could potentially provide a more credible representation of the historical condition when compared to the recent decade with record-high warming and extremes. In addition to reporting the relative change in occurrence, providing information on the actual change and discussing the possible influence of uncertainties would be helpful for a better understanding of their findings.

Thank you for these comments. We acknowledge the limitation in validation assessment due to the very limited number of lakes with in-situ measurements available. Despite this, the widespread distribution of lakes globally and the substantial amount of matching data give us confidence in the three published dataset's ability to depict the three types of extreme events effectively. The only thing that we could do in the future is to continue collecting more observational data to validate these global-scale datasets more thoroughly.

Regarding the uncertainty in historical baseline results due to the low temporal resolution of the global algal bloom dataset, we firstly want to clarify only historic algal bloom frequency based on Hou's global algal bloom dataset faces such uncertainty, since other two dataset provide seamless monthly or daily data. Hou et al. (2022) conducted detailed quality control steps to ensure the reliability of the frequency calculations for the historic period. Specifically, a minimum valid observation of 45 during the historical period was used in the calculation of occurrence of algal bloom data, as presented in Hou et al. (2022). Therefore, we are very confident and convinced that the calculated frequency results are valid. Meanwhile, to address the reviewer's concern, we now also made a sensitivity analysis by using high temporal resolution algal bloom data derived from another satellite product (Aqua MODIS with daily temporal resolution; Wang et al., 2023). Consistent result with different proportion of observations can further support our confidence in the historical baseline calculations and frequency changes (see Fig. R1). The figure and analysis are now added in the manuscript (Supplementary Fig. S11).

Lines 648-662: Despite the differences in observation frequency, our analysis revealed that the overall change direction (increase or decrease) of algal blooms in Lake Taihu based on partial or all-available hourly-resolution GOCI images (Qi et al., 2018) remained unchanged (Fig. S10). Using another high-resolution satellite, Aqua MODIS, derived daily algal bloom dataset for 102 Chinese lakes (Wang et al., 2023), we performed frequency calculation for historic period and frequency

change calculation during historic and contemporary periods. For the historic period, only a partial proportion of the satellite observations were used. The results demonstrate that the frequency changes calculated from partial observations are closely consistent with those obtained using all data (Fig. S11a). We further calculated the standard deviation of the differences between the two across all lakes, representing the uncertainty propagated into the historic frequency due to gaps in remote sensing observations (0.066%, Fig. S11b). By excluding lakes where the algal bloom frequency less than this uncertainty threshold, we ensured the reliability of our reported frequency change of extremes. This suggests that while observation frequency is an important factor in detecting extreme events, the reported increase in lake extremes observed in our study is not an artifact of differences in observation frequency. This finding underscores the significance of the reported increase in lake extremes and its resilience to potential variations in observation frequency. However, it is important to recognize that observation frequency can influence the detection and characterization of extreme events, and future research may benefit from utilizing higher-frequency observation data to provide a more detailed understanding of extreme event dynamics, which would also aid in improving our understanding of compounding events, where two or more extremes occur sequentially. Nonetheless, our study provides valuable insights into the changing patterns of lake extremes over time, even within the constraints of varying observation frequency.

Fig. R1 | The influence of MODIS observation quantity on algal bloom frequency change during the historic period of 2003-2012 and present period of 2018-2022. Shown is (a) comparison of algal bloom frequency change detected using all available MODIS observations and using a specific proportion of the original data for historical period frequency calculations. Different colors represent varying percentage of observations used. A zoomed-in panel highlights the details of their differences. Also shown is (b) the histogram of the difference between the two frequency changes, with mean (-0.006%) and standard deviation (i.e., uncertainty, 0.066%) annotated. Frequency is calculated based on daily algal bloom area data for 102 lakes in China from 2003-2022 (Wang et al., 2023). For each lake, 10 choices of proportion were conducted.

Using the high-resolution MODIS derived daily algal bloom data from 102 Chinese lakes, we calculated an uncertainty of 0.066% (Fig. R1) arising from the temporal resolution of the satellite data. In turn, we have updated our analysis to only include lakes with an algal bloom frequency of more than 0.066% during the historic period. Whilst this considerably reduces the number of studied sites, the summary statistics throughout the manuscript are only marginally different.

We did not represent results of the actual occurrence values of extreme events in the manuscript primarily because this is a key result of Hou’s work and our work is to analyze the decadal changes in three types of extreme events in lakes. To address the reviewer’s concern, we present the histogram of algal bloom frequency values of the studied lakes for the historic period and present period here. As shown in Fig. R2, the average frequency was 2.32% and 3.98% for the two focused periods.

Fig. R2 | Histogram of algal bloom frequency for historic period (1980s to 1999) and contemporary period (2010s). The mean and max frequency values of the studied lakes are also annotated.

Regarding the exclusion of the period of the 2000s, we recognize that it might provide a more credible historical baseline, especially considering the recent decade's record-high warming and extremes. However, we do not think that it would be appropriate to compare data from 1980-2010 (approx. 30 years) with 2010-2020 (approx. 10 years) in this study. We also do not think that it would be appropriate to solely compare the period of 2000-2010 with 2010-2020 as the study period would then be approx. 20 years, which seems too short to quantify a climatic/anthropogenic response (i.e., if data is available, one often uses a period of at least 30 years to identify a climate trend).

We are not entirely sure what the reviewer is referring to in terms of “actual occurrence values” of extreme events. We estimated the frequency of extreme events in each lake; this was chosen to ensure that the sampling frequency did not bias our analysis. Thus, here we report the actual changes in these frequencies, which have units of percentage (i.e., the values are not reported as a % change).

Reference:

Wang, Y., Feng, L., & Hou, X. (2023). Algal blooms in lakes in China over the past two decades: Patterns, trends, and drivers. *Water Resources Research*, 59(10), e2022WR033340.

The current analysis for identifying drivers is highly uncertain, particularly for bivariate and multivariate extremes. Most of descriptions regarding the drivers in the Results section are discussion-oriented. I would recommend move this part to the Discussion. The current Discussion section focuses on the general implications of single or multivariate extremes rather than delving into the specific findings of this study.

The reviewer’s suggestion to move the discussion of drivers from the Results section to the Discussion is noted, and we agree that this adjustment will enhance the clarity of our presentation. In response to their suggestion, we have relocated the discussion of drivers to the Discussion section, providing a more cohesive and focused narrative. We also ensure that the Discussion section

explicitly addresses the specific findings of our study, delving into the nuances of both single and multivariate extremes, as rightly pointed out.

Reviewer #2

The paper describes the changes globally in three parameters by comparing two decadal periods. Importantly they analyse the concordance of changes globally identifying areas where multiple pressures are acting on lakes. This is a most welcome move to more holistic studies on the impact of climate change. The paper is of high standard and adequate corrections have been made where requested. The closing paragraph of the discussion is of exceptional quality.

I recommend publication with minor corrections. There are some comments and minor issues listed below which the author / editor can consider for attention.

Thank you once again for reviewing our paper. We very much appreciate your time and your efforts to help us improve the paper. We have addressed all comments below.

Line 23: Consider rephrasing first sentence for a stronger start. **Done**

Line 397 change One notably concern - to One notable concern **Done**

Line 453 change cantered to centered **Done**

Figure 1. If not too difficult, consideration should be given to changing the background world map colour to something other than grey. Alternatively, remove grey as an option in the colour legend.

Done. Good point.

Reviewers' comments:

Reviewer #1 (Remarks to the Author):

I appreciate the efforts made by the authors including improved wording and one additional validation. However, major issues persist in both the level of significance and the quality of presentation, which may affect its suitability for publication in Nature Communications.

The two major conclusions of this study are both based on the percentage of lakes. The authors studied three categories of extreme events for 2,724 lakes using data from their previously published data on each type of extreme event, derived on satellite observations. It is worth noting that this sample only represents a very small fraction of the global lakes investigated in their previous studies, accounting for 3%, 1%, and 0.2% of lakes in Tong et al., Hou et al., and Zhao et al., respectively. While the exploration of multivariate lake extremes is novel, this study falls short in the broad significance considering two key aspects. First, it does not present a comprehensive global dataset on lakes that can be widely applied in future studies. Second, due to the absence of strategic sampling, the knowledge generated from the studied lakes might not be readily transferable to other groups of lakes or the broader global conditions of lakes, as noted by the authors. As a result, the significance of this study may be limited to readership within a more specialized journal.

I comment that the authors added the third category, labeled 'stable', to describe minor changes in lake extremes. This addition appears to enhance the clarity of this manuscript in certain areas, but the continued use of the old binary classification in many places introduces additional confusions. According to their new analysis, excluding lakes with minor changes led to a decrease in the reported percentage of lakes experiencing increasing algal blooms from 56% to 29%. However, the reported percentages of lakes in the abstract and all figures in the main narrative remain unchanged, relying on the binary classification. Specifically, in the abstract, the authors still report that 75% of global lakes experienced a concurrent increase in two or more extremes without considering whether these increases are minor or subject to uncertainties.

Line 33: "The greatest increases in the frequency of these extremes was found in regions". Please change "was" to "were".

Line 101: "if their frequency changed by $>+1\%$ ". Please clarify whether this threshold is a relative percentage or an absolute change. The authors may explain why this threshold was chosen, same comments for other two categories of extremes.

Line 721 No statements regarding data availability of the generated datasets are provided.

Reviewer #1 (Remarks to the Author):

I appreciate the efforts made by the authors including improved wording and one additional validation. However, major issues persist in both the level of significance and the quality of presentation, which may affect its suitability for publication in Nature Communications.

We sincerely thank the reviewer for their thoughtful review and constructive feedback. Their insights have been valuable in strengthening our manuscript. We have carefully considered all suggestions and incorporated many of them throughout the manuscript (including revisions made in previous rounds). For specific points where we have retained our original approach, we have provided detailed explanations in the response document to justify our decisions. We believe in open and respectful discourse, and we hope that the reviewer will appreciate the rationale behind our choices, even if they differ from their initial perspective. We thank the reviewer for their contribution.

In the following sections, we have provided point-by-point responses regarding your thoughtful comments and constructive suggestions. We also have made corresponding changes in the revised manuscript. We hope that we could address all issues suitably and would have potential opportunity to proceed with our manuscript. Thanks again for your time and effort.

The two major conclusions of this study are both based on the percentage of lakes. The authors studied three categories of extreme events for 2,724 lakes using data from their previously published data on each type of extreme event, derived on satellite observations. It is worth noting that this sample only represents a very small fraction of the global lakes investigated in their previous studies, accounting for 3%, 1%, and 0.2% of lakes in Tong et al., Hou et al., and Zhao et al., respectively. While the exploration of multivariate lake extremes is novel, this study falls short in the broad significance considering two key aspects. First, it does not present a comprehensive global dataset on lakes that can be widely applied in future studies. Second, due to the absence of strategic sampling, the knowledge generated from the studied lakes might not be readily transferable to other groups of lakes or the broader global conditions of lakes, as noted by the authors. As a result, the significance of this study may be limited to readership within a more specialized journal.

Thanks for your valuable comment. We must admit the studied lakes is only a small part of global lakes presented in the three datasets. To be more specific, lakes are included when there is a bloom event detected, namely a total number of 21,878 bloom-affected lakes detected in Hou et al.'s study. We can only calculate bloom occurrence change for these bloom-affected lakes. So, these bloom-affected lakes could be the actual population for the global scale study.

However, the final samples in this study are only 2,724 lakes limited by the two steps of selections. The first one is limitation of GLAST dataset, which is primarily limited by the moderate spatial-resolution (60-120 m) of thermal infrared bands, which can pollute lake temperature signals when the lake center is very close to land. Among all 92,245 lakes included in GLAST dataset, only 4,530 lakes are detected as bloom-affected [*This is the first step of sampling*]. This is the number of our studied lakes in the original manuscript. We are very appreciated that reviewer pointed out the uncertainty that could be propagated into the historic frequency due to gaps in remote sensing observations. We calculated the uncertainty and removed those lakes with frequency change less than this uncertainty threshold to ensure the reliability of our reported frequency change of extremes [*This is the second step of sampling*]. The determination of studied lakes in this study has been summarized in Fig. R1a below.

We acknowledge that our lake sampling was entirely non-strategic, chosen due to limitations in the available data. To validate the representativeness of the selected lakes on a global scale, we conducted statistical comparisons of latitude distribution, longitude distribution, and climatic zone distribution based on both the number and area of lakes. We compared the distribution consistency between the lakes studied in this paper (2,724 lakes) and the global bloom-affected lakes (21,878 lakes). As shown in Fig. R1b-h, the relatively consistent spatial coverage and distribution in different aspects could support the potential of the studied lakes to represent bloom-affected lakes globally.

In conclusion, the studied lakes encompass a diverse range of lakes in terms of geolocational distribution and climate zones. Therefore, we believe that our study findings could be readily applicable to a wider range of global bloom-affected lakes.

We have added the information in the revised manuscript (lines 466-468):

These lakes span various latitudes and longitudes (Fig. S5c-f), and their distribution across different climate types closely mirrors that of global eutrophic lakes (Fig. S5g-hf).

Fig. R1 | Representativeness of examined lakes compared to global bloom-affected lakes. Shown is (a) the determination process of the studied lakes, and (b) their spatial distribution as well as the (c-d)

longitudinal profiles and (e-f) latitudinal profiles of lake numbers (c,e) and lake areas (d,f). In panels (b-f), black dots or curves represent global bloom-affected lakes, and red dots or curves represent the studied lakes. Also shown is (g-h) comparison of the percentage (%) of lakes (g, number; h, area) in various climatic zones between the studied lakes and global eutrophic lakes. Climate zones are classified according to the Köppen–Geiger world map of climate classification (Kottek et al., 2006; <http://koeppen-geiger.vu-wien.ac.at/present.htm>). The annotations of ‘A*’, ‘B*’, ‘C*’, ‘D*’, and ‘E*’ in the legend of panels (g) and (h) represent tropical, arid, temperate, cold, and polar climate zones, respectively.

Reference:

Kottek, M., Grieser, J., Beck, C., Rudolf, B., & Rubel, F. (2006). World map of the Köppen-Geiger climate classification updated.

Finally, we do not agree that the significance of this study may be limited to readership within a more specialized journal, strictly due to the number of study sites. There are numerous other high impact papers that have been published in Nature journals in very recent years that have similar or fewer number of study sites. For example,

Jane, S.F., Hansen, G.J.A., Kraemer, B.M. et al. Widespread deoxygenation of temperate lakes. *Nature* 594, 66–70 (2021) – investigated 393 lakes.

Kraemer, B.M., Pilla, R.M., Woolway, R.I. et al. Climate change drives widespread shifts in lake thermal habitat. *Nat. Clim. Chang.* 11, 521–529 (2021) – investigated 139 lakes.

Yao, F. et al. Satellites reveal widespread decline in global lake water storage. *Science* 380, 743-749 (2023) – investigated 1972 lakes.

In brief, our sample of lakes is (often) much greater than that included in other high impact papers. The novelty of this study is in terms of studying multivariate extremes, and not in terms of the number of sites studied – we don’t believe that the number of sites determines its broad significance.

I comment that the authors added the third category, labeled ‘stable’, to describe minor changes in lake extremes. This addition appears to enhance the clarity of this manuscript in certain areas, but the continued use of the old binary classification in many places introduces additional confusions. According to their new analysis, excluding lakes with minor changes led to a decrease in the reported percentage of lakes experiencing increasing algal blooms from 56% to 29%. However, the reported percentages of lakes in the abstract and all figures in the main narrative remain unchanged, relying on the binary classification. Specifically, in the abstract, the authors still report that 75% of global lakes experienced a concurrent increase in two or more extremes without considering whether these increases are minor or subject to uncertainties.

We greatly appreciate this comment. Before we respond to the specific comment, we would like to highlight that despite the numbers that we quote for lakes globally, regional scale patterns of change can also be substantial - of notable concern is the changes reported for India, with a growing population (18% of the global population), and a dramatic increase in extreme events in freshwater lakes.

Firstly, we sincerely apologize for overlooking this valuable revision in the abstract and other critical

sections of the manuscript, in alignment with the new classification of change types. Secondly, we extend our sincere apologies for previously relying solely on a threshold value of $\pm 1\%$, based on our research experience, to categorize the change types. We now recognize the necessity of employing a statistically quantitative method to establish a threshold for distinguishing between significant change and minor change.

Regarding the selection of threshold values, we conducted a series of quantitative analyses. Specifically, concerning lake heatwaves and lake low water extremes, we leveraged the high temporal resolution of these datasets. Initially, we computed the annual frequency of extremes. Subsequently, we conducted a significance test for the annual trend changes, wherein a P -value greater than 0.05 indicated minor changes of period frequency change.

In the case of lake bloom extremes, conducting the same statistical analyses was not feasible due to limitations in the dataset provided by Hou's global algal bloom dataset. This dataset only offers period-calculated frequency of extremes and lacks a time series of algal bloom occurrence data, thus preventing us from calculating a P -value for the temporal change. To address this limitation, we utilized the MODIS-derived daily algal bloom dataset for 102 Chinese lakes (Wang et al., 2023). In detail, we initially computed the relative algal bloom frequency change during the historic and present periods, along with the annual trend and corresponding P -value based on annually bloom frequency data. Subsequently, we classified the relative change of bloom frequency into two groups: one exhibiting a significant temporal change pattern (P -value ≤ 0.05), and another showing an insignificant temporal change pattern (P -value > 0.05). As illustrated in Fig. R2, we observed a distinct threshold value (0.4) of relative change of bloom frequency that effectively differentiated the two groups. While there were some overlaps between the two groups, we consider this method as a viable alternative for quantitatively determining the specific threshold of bloom frequency change to identify lakes with "notable" change.

Based on the above quantitative thresholds for three extremes, we subsequently classified the studied lakes into distinct categories: i.e., those exhibiting notable change. We meticulously recomputed all outcomes and integrated the new findings into the abstract and various sections throughout the manuscript, rectifying our previous oversight. The revisions made are enumerated below:

Lines 30-33:

Our study, which focuses on bloom-affected lakes, suggests that 75% (27%) of the studied sites have experienced a concurrent (notable) increase in at least two of the extreme events considered, with 25% (5%) experiencing an (notable) increase in frequency of all three extremes.

Lines 103-105:

We calculate that 40% of the studied lakes experienced a notable increase in algal blooms frequency, 26% experienced a notable decrease.

Lines 137-138:

We calculate that 46% of the studied lakes experienced a notable increase, 1% experienced a notable decrease.

Lines 167-169:

We calculate that 18% of the studied lakes experienced a notable increase, 20% experienced a notable decrease.

Lines 215-219:

Overall, our study suggests that approximately three quarters of the studied lakes ($n = 2,035$; 75%) have experienced a simultaneous increase (27% experiencing a notable increase) in at least one of these bivariate extreme events during the study period, with some (see below) experiencing an increase in all three extremes.

Lines 221-223:

Our analysis suggests that 51% of the studied lakes ($n = 1,398$) have experienced a concurrent increase (20% experiencing a notable increase and 0% a notable decrease) in the frequency of algal blooms and lake heatwaves (Fig. 2).

Lines 237-239:

Regarding the second bivariate extreme of interest, we estimate that 46% ($n = 1,266$) of the studied lakes have experienced a simultaneous increase (9.5% experiencing a notable increase and 0% a notable decrease) in the frequency of lake heatwaves and low water extremes (Fig. 2).

Lines 249-251:

Most notably, 27% of the studied lakes experienced an increase (8% with a notable increase and 3.8% with a notable decrease) in the frequency of this bivariate extreme event (Fig. 2).

Lines 264-266:

We calculate that approximately one-quarter of the studied lakes ($n = 687$; 25%) have experienced a simultaneous increase (5% with a notable increase and 0% with a notable decrease) in all three univariate extremes since the 1980s (Fig. 3).

Lines 707-719:

For extreme temperature events and water level extremes, we determined significance (or stability) of occurrence changes using statistical test P -values of annual trend changes, leveraging the high-temporal resolution of GLAST and GLEV datasets. Due to limitations of the GBD dataset, we used an alternative method to determine frequency change thresholds corresponding to statistically significant levels. Utilizing the MODIS-derived daily algal bloom dataset for 102 Chinese lakes (Wang et al., 2023), we computed relative algal bloom frequency changes between historic and present periods, alongside annual trends and P -values. We then grouped relative bloom frequency changes into two categories: significant temporal change (P -value ≤ 0.05) and insignificant temporal change (P -value > 0.05). As shown in Fig. S13, this approach revealed a distinct threshold value (0.4) effectively distinguishing between the two groups. Despite some overlap, we find this method a viable alternative for quantitatively identifying 'notable' change in bloom frequency.

Fig. R2| Comparison of relative changes in lake algal bloom frequency between significantly and insignificantly changing long-term trends. Shown are the (a) histogram and (b) boxplot depicting the comparison between the two groups.

However, please note that we still include the original binary classification in the paper, as it provides important information. This binary classification also describes if the changes are statistically significant (via the KS-test).

Line 33: “The greatest increases in the frequency of these extremes was found in regions”. Please change “was” to “were”.

Thank you for bringing this error to our attention. We have made the necessary revision accordingly.

Line 101: “if their frequency changed by >+1%”. Please clarify whether this threshold is a relative percentage or an absolute change. The authors may explain why this threshold was chosen, same comments for other two categories of extremes.

Thank you for your valuable comment. Initially, the threshold referred to the absolute change in extreme frequency between the historical and present periods. Since the frequency for both periods share the same unit (i.e., %), the difference (i.e., change) between the two periods is also expressed in %. This may lead to confusion between absolute and relative change. To address this concern and provide clarity, we have included the specific information for the new determined method description in the revised manuscript.

In the previous version, we determined the threshold solely based on our research experience and personal judgment. Initially, we believed that even a 1% increase in extremes warranted attention because they exceeded normal levels, and any additional increase could be deemed significant. However, this approach lacked scientific justification. To rectify this, as outlined in major comment 1, we employed statistical testing methods to determine the threshold to classify notable change and notable change for the three extremes. We sincerely appreciate your valuable and consistent emphasis on this critical point.

Line 721 No statements regarding data availability of the generated datasets are provided.

Thanks for this comment. We have added the data availability statement in the revised manuscript.

REVIEWERS' COMMENTS

Reviewer #2 (Remarks to the Author):

Thanks for the opportunity to review the manuscript. This is the third revision, it is an excellent paper and the global patterns and regional concordance of some 'perfect storm' trends are perfect for publication in nature. I think the remaining suggested corrections have been addressed well by the authors.

In particular:

- 1) It has been clarified that lower numbers of lakes were included as only bloom lakes considered. These were further stratified so that temperature estimates from satellite would not be contaminated by edge effect of smaller lakes – perfectly standard and necessary in the use of remote sensing data.
- 2) Carried out analysis to show that their data is representative of larger dataset on lake blooms.
- 3) Re-evaluated the thresholds for change in a satisfactory manner. Inadvertently this examination of the threshold for changing bloom patterns will be extremely useful to practitioners.

I recommend publication following addressing of the minor wording issues below:

Abstract text: lines 30 to 37 (need to alter the abstract regarding the wording of notable – a suggestion below)

Our study, which focuses on bloom-affected lakes, suggests that 75% (27%) of the studied sites have experienced a concurrent (notable) increase in at least two of the extreme events considered (with 27% defined as having a notable increase), with 25% (5%) experiencing an (notable) increase in frequency of all three extremes (5% had a notable increase). The greatest increases in the frequency of these extremes were found in regions that have experienced parallel increases in agricultural fertilizer use, mean lake warming, and a decline in water availability. As extreme events in lakes become more common, understanding their impacts must be a primary focus of future studies and they must be carefully considered in future risk assessments.

Line 100 Moreover, in this study, we classified the studied lakes as experiencing a “notable” (or considerable) increase or decrease in the occurrence of algal blooms if their decadal change in frequency surpassed a predetermined statistically significant level (see Methods).

I think it is more appropriate to refer to this as something like the below:

a “notable” (or considerable) increase or decrease in the occurrence of algal blooms if their decadal change in frequency surpassed a 0.4 increase in relative frequency (that closely aligned with a statistically significant change (see Methods)).

Line 140 “a notable increase (i.e., that surpassed a predetermined statistically significant level)”

This leaves the reader wondering a bit - consider providing a numeric figure for this e.g. surpassed a 0.4

increase in relative frequency (see methods).

Line 172 same as above

Reviewer #2

Thanks for the opportunity to review the manuscript. This is the third revision, it is an excellent paper and the global patterns and regional concordance of some 'perfect storm' trends are perfect for publication in nature. I think the remaining suggested corrections have been addressed well by the authors.

In particular:

- 1) It has been clarified that lower numbers of lakes were included as only bloom lakes considered. These were further stratified so that temperature estimates from satellite would not be contaminated by edge effect of smaller lakes – perfectly standard and necessary in the use of remote sensing data.
- 2) Carried out analysis to show that their data is representative of larger dataset on lake blooms.
- 3) Re-evaluated the thresholds for change in a satisfactory manner. Inadvertently this examination of the threshold for changing bloom patterns will be extremity useful to practitioners.

I recommend publication following addressing of the minor wording issues below:

Thank you very much for reviewing our paper and for your suggestions through each round of review.

Abstract text: lines 30 to 37 (need to alter the abstract regarding the wording of notable – a suggestion below)

Our study, which focuses on bloom-affected lakes, suggests that 75% (27%) of the studied sites have experienced a concurrent (notable) increase in at least two of the extreme events considered (with 27% defined as having a notable increase), with 25% (5%) experiencing an (notable) increase in frequency of all three extremes (5% had a notable increase). The greatest increases in the frequency of these extremes were found in regions that have experienced parallel increases in agricultural fertilizer use, mean lake warming, and a decline in water availability. As extreme events in lakes become more common, understanding their impacts must be a primary focus of future studies and they must be carefully considered in future risk assessments.

Thank you. We have revised the abstract accordingly.

Line 100 Moreover, in this study, we classified the studied lakes as experiencing a “notable” (or considerable) increase or decrease in the occurrence of algal blooms if their decadal change in frequency surpassed a predetermined statistically significant level (see Methods).

I think it is more appropriate to refer to this as something like the below:

a “notable” (or considerable) increase or decrease in the occurrence of algal blooms if their decadal change in frequency surpassed a 0.4 increase in relative frequency (that closely aligned with a statistically significant change (see Methods)).

Changed

Line 140 “a notable increase (i.e., that surpassed a predetermined statistically significant level)”

This leaves the reader wondering a bit - consider providing a numeric figure for this e.g. surpassed a 0.4 increase in relative frequency (see methods).

Done

Line 172 same as above

Done